# Stromal androgen signaling acts as tumor niches to drive prostatic basal epithelial progenitor-initiated oncogenesis

Alex Hiroto[1,4], Won Kyung Kim[1,4], Ariana Pineda[1], Yongfeng He[1], Dong-Hoon Lee[1], Vien Le[1], Adam W. Olson[1], Joseph Aldahl[1], Christian H. Nenninger[1], Alyssa J. Buckley[1], Guang-Qian Xiao[2], Joseph Geradts[3] & Zijie Sun [1] ✉

The androgen receptor (AR)-signaling pathways are essential for prostate tumorigenesis. Although significant effort has been devoted to directly targeting AR-expressing tumor cells, these therapies failed in most prostate cancer patients. Here, we demonstrate that loss of AR in stromal sonic-hedgehog Gli1-lineage cells diminishes prostate epithelial oncogenesis and tumor development using in vivo assays and mouse models. Single-cell RNA sequencing and other analyses identified a robust increase of insulin-like growth factor (IGF) binding protein 3 expression in AR-deficient stroma through attenuation of AR suppression on Sp1-regulated transcription, which further inhibits IGF1-induced Wnt/β-catenin activation in adjacent basal epithelial cells and represses their oncogenic growth and tumor development. Epithelial organoids from stromal AR-deficient mice can regain IGF1-induced oncogenic growth. Loss of human prostate tumor basal cell signatures reveals in basal cells of stromal AR-deficient mice. These data demonstrate a distinct mechanism for prostate tumorigenesis and implicate co-targeting stromal and epithelial AR-signaling for prostate cancer.

The prostate is an androgen-regulated organ[1]. The activation of the androgen receptor (AR) through direct binding of androgens contributes to prostate tumorigenesis[2,3]. Over the past decades, significant effort has been devoted to determining the intrinsic mechanism underlying AR action in prostate tumor cells[4–6]. Androgen deprivation therapy (ADT) directly targeting AR-expressing prostate tumor cells fails in most prostate cancer patients who consequently develop castration-resistant prostate cancer (CRPC), an incurable disease[2,3]. Therefore, the role of AR in prostate tumorigenesis needs to be re-evaluated for designing effective therapeutic strategies.

Emerging evidence has shown that the dynamic interactions of prostate cancer cells with their surrounding stroma are essential in tumor initiation, progression, and hormone refractoriness[7–10]. The AR is also expressed in prostate stromal cells[11–13]. However, the regulatory role of stromal AR in facilitating prostate epithelial oncogenesis remains largely unknown. Additionally, the presence and identity of prostate stromal cells acting as a tumor niche are also unclear. An indispensable role of stromal AR action in sonic hedgehog (Shh) responsive Gli1-lineage cells in prostate development, pubertal growth, and regeneration has been reported[14], implicating the critical role of paracrine interactions between prostatic epithelia and stroma during prostate development and growth.

In this study, using in vivo tissue recombination assays and newly generated mouse models, we examined if stromal AR signaling in Gli1-lineage cells acts as tumor niches in prostate oncogenesis. We demonstrate that loss of AR in stromal Shh-responsive Gli1-lineage cells diminishes prostate epithelial oncogenesis and tumor development. Single-cell RNA sequencing analyses and other experimental

[1]Department of Cancer Biology, Cancer Center and Beckman Research Institute, City of Hope, Duarte, CA, USA. [2]Department of Pathology, Keck School of Medicine, University of Southern California, Los Angeles, CA, USA. [3]Department of Pathology and Laboratory Medicine, Brody School of Medicine, East Carolina University, Greenville, NC, USA. [4]These authors contributed equally: Alex Hiroto, Won Kyung Kim. ✉e-mail: zjsun@coh.org

approaches identify robust increased expression of insulin-like growth factor binding protein 3 (IGFBP3) in AR-deficient Gli1-lineage stromal cells through attenuation of AR suppression on Sp1-regulated transcription. Aberrantly increased IGFBP3 impairs IGF1-induced Wnt signaling activation in subsets of basal epithelial cells, directly inhibiting prostatic epithelial oncogenic growth and tumor development. Incubation of IGFBP3 with IGF1 also showed to significantly attenuate IGF1-induced oncogenic growth of prostatic organoids derived from mouse basal epithelia. Loss of human prostate tumor basal cell signatures was identified in basal cells of stromal AR-deficient mice. These data demonstrate the critical role of stromal AR in Gli1-lineage cells, acting as a tumor niche to support prostate epithelial tumorigenesis and implicate the underlying mechanisms for hormone refractoriness and CRPC development through dysregulation of stromal AR on IGFBP3-IGF1 signaling during ADT therapies. Therefore, further investigation of therapeutic strategies by co-targeting reciprocal interactions of AR and IGF1 pathways between epithelial tumor cells and surrounding tumor niches may improve clinical outcomes for advanced prostate cancer.

## Results

### Loss of stromal AR in Gli1-lineage cells diminishes prostatic epithelial oncogenesis

Using in vivo tissue recombination assays, we first examined if stromal AR action in Gli1-lineage cells acts as a tumor niche to support prostate epithelial oncogenesis[15]. The urogenital sinus mesenchyme (UGM) of either $Ar^{L/Y}$:$Gli1^{CreER/+}$ (ARKO) mice[14], with tamoxifen (TM) induced $Ar$ deletion in Gli1-expressing cells, or $Gli1^{CreER/+}$ controls, and the urogenital sinus epithelium (UGE) of $Ctnnb1^{L(ex3)/+}$:$PB^{Cre4}$ mice, a prostate cancer model, with stabilized β-catenin expressed in prostate epithelium were isolated at E16.5, and transplanted together under the renal capsule of NOD/SCID mice (Fig. 1a). Gross analyses after 12-week implantation showed noticeable impaired growth in grafts combined with AR-deficient UGM and $Ctnnb1^{L(ex3)/+}$:$PB^{Cre4}$ UGE in comparison to

counterparts with control UGM, which were transparent and significantly heavier ($p = 0.012$) and larger than the ARKO grafts (left panel, Fig. 1b, c, d vs 1h, i). Pathological lesions resembling typical mouse prostatic intraepithelial neoplasia 3–4 (mPIN3 to 4)[16] appeared in the control UGM combined grafts (Fig. 1e). Immunohistochemical analyses (IHC) showed positive nuclear and cytoplasmic β-catenin in atypical cells within PIN lesions, implicating the oncogenic role of stabilized β-catenin in PIN formation (red arrows, Fig. 1f). Positive nuclear AR staining appeared in both atypical epithelial and stromal cells (pink and blue arrows, respectively, Fig. 1g). In contrast, grafts derived from ARKO UGM with $Ctnnb1^{L(ex3)/+}$:$PB^{Cre4}$ UGE showed mild pathological lesions resembling prostatic hyperplasia and PIN1 (Fig. 1j). Accordingly, very few cells showed positive β-catenin nuclear staining ($p < 0.001$; red arrow, Fig. 1k; Supplementary Fig. 1a). Positive nuclear AR staining mainly appeared in epithelial cells but was lacking in stromal cells (pink arrows, Fig. 1l). Significantly fewer Ki67+ cells ($p = 0.001$) appeared in the above grafts than those derived from control UGM (right panel, Fig. 1b). In this study, we also performed tissue recombination assays using AR-deficient and control UGMs as described above with UGEs isolated from another prostate cancer mouse model, $Pten^{L/L}$:$PB^{Cre4}$ mice[17], and observed mild pathologic changes and impaired oncogenic growth in the samples recombined with AR-deficient UGMs (Supplementary Fig. 1b, c). Altogether, these data demonstrate a promotional role of stromal AR signaling in Gli1-expressing cells to support prostate epithelial oncogenesis and tumor initiation.

### Deletion of stromal AR in Gli1-lineage cells impairs prostatic epithelial oncogenesis and tumor development

To evaluate intrinsic stromal AR signaling in prostatic epithelial oncogenesis, we developed $Hi$-$Myc$:$Ar^{L/Y}$:$Gli1^{CreER/+}$ mice, in which the $human\ c$-$Myc$ transgene ($hMycTg$) expression controlled by the modified probasin promoter in prostate epithelium[18] co-occurs with stromal $Ar$ deletion regulated by Gli1-driven $CreER$ in prostate stroma[14]

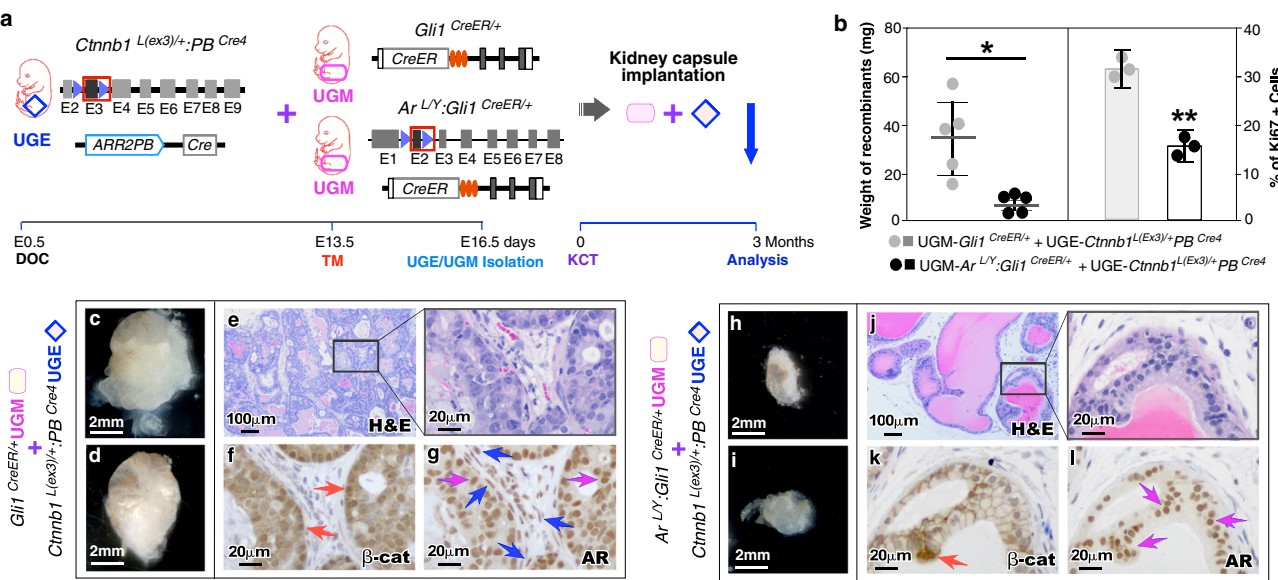

**Fig. 1 | Deletion of AR in Gli1-expressing cells in the UGM impairs UGE oncogenic xenograft growth. a** Schematic of the experimental timeline for tissue recombination and kidney capsule transplantation. Urogenital sinus epithelium (UGE) from $Ctnnb1^{L(ex3)/+}PB^{Cre4}$ embryos and urogenital sinus mesenchyme (UGM) from $Gli1^{CreER/+}$ or $Ar^{L/Y}$:$Gli1^{CreER/+}$ embryos after activation of $Gli1^{CreER/+}$ are recombined and grown under the renal capsule of host mice. DOC, day of conception; TM, tamoxifen; KCT, kidney capsule transplantation. **b** Weights of xenograft recombinants from the indicated UGE and UGM combinations (left panel). Quantification of the percentage of Ki67+ cells per total cells (right panel). Data are represented as

mean ± SD of five biological replicates. Two-sided Student's *t* test, *$p < 0.05$, **$p < 0.01$. Source data and the exact *p*-values are provided as a Source Data file. **c–l** Representative gross (**c–d, h–i**), hematoxylin-eosin (H&E) (**e, j**), and immunohistochemistry (IHC) for β-catenin (β-cat) and androgen receptor (AR) (**f–g, k–l**) images of xenograft tissues from the indicated UGE and UGM combinations. Scale bars, 2 mm (**c–d, h–i**), 100 μm (**e, j**), and 20 μm (**f–g, k–l**). The nuclear expression of β-cat in the epithelium is indicated by red arrows. Pink and blue arrows indicate epithelial and stromal AR+ cells, respectively.

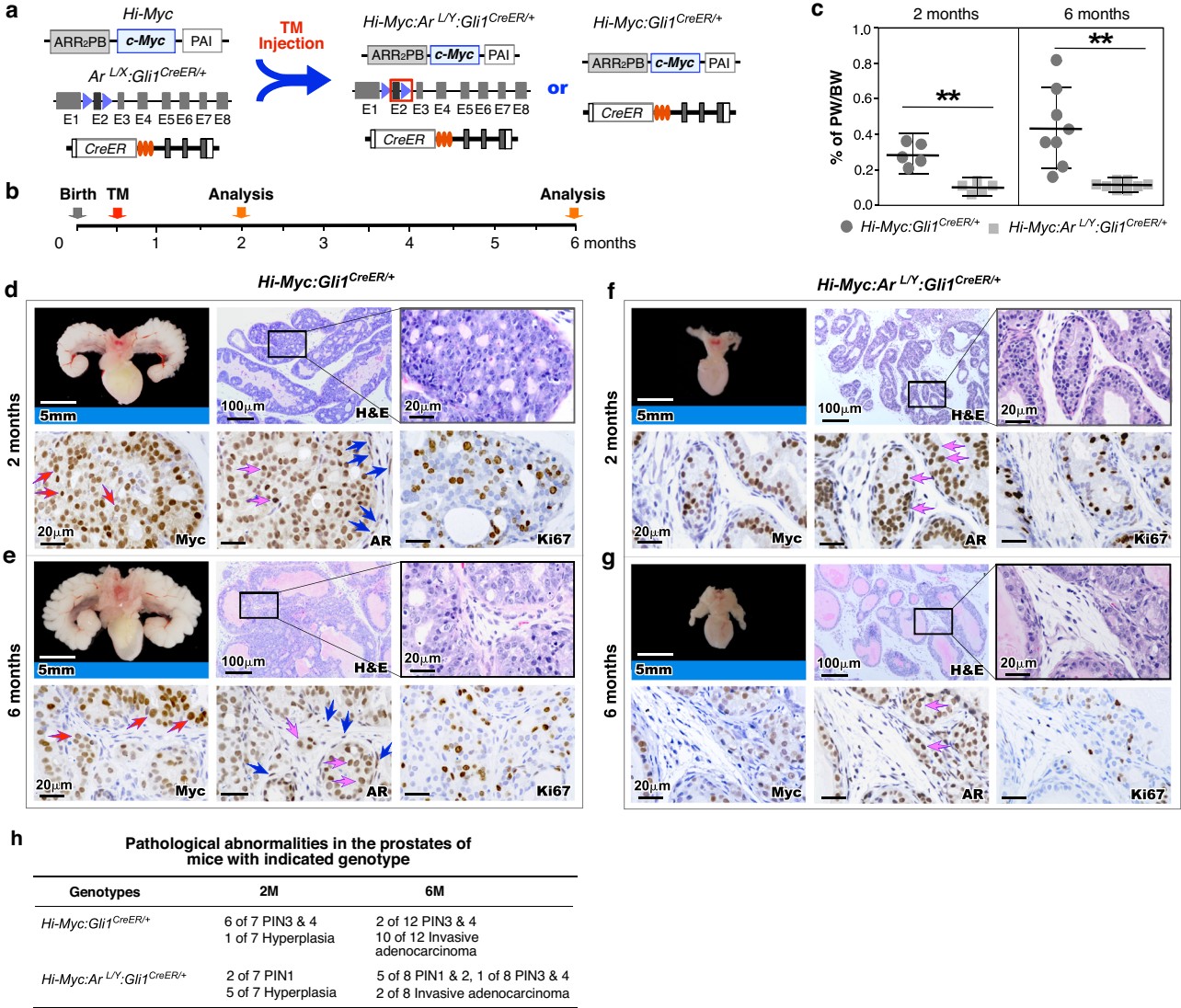

**Fig. 2 | Deletion of AR in Gli1-expressing cells impairs prostate epithelial oncogenic transformation and growth in vivo. a** Schematic for generating *Hi-Myc:Gli1*[CreER/+] and *Hi-Myc:Ar*[L/Y]:*Gli1*[CreER/+] mice. **b** Schematic of experimental timeline for tamoxifen (TM) injection and analysis. **c** Ratio of prostate weight (PW) versus whole body weight (BW) as percentages of 2-month-old mice (left panel; *n* = 5 biological replicates) and 6-month-old mice (right panel; *n* = 8 biological replicates). Data are represented as mean ± SD of indicated biological replicates. Two-sided Student's *t* test, **p < 0.01. Source data and the exact *p* values are provided as a Source Data file. **d**–**g** Representative images of gross urogenital tissues, H&E, and IHC staining for Myc, AR, and Ki67 in adjacent prostate tissue sections of *Hi-Myc:-Gli1*[CreER/+] and *Hi-Myc:Ar*[L/Y]:*Gli1*[CreER/+] mice. Red arrows indicate nuclear Myc+ cells. Epithelial and stromal AR+ cells are indicated by pink and blue arrows, respectively. **h** Table summarizing the pathological abnormalities in the prostates of 2- and 6-month-old *Hi-Myc:Gli1*[CreER/+] and *Hi-Myc:Ar*[L/Y]:*Gli1*[CreER/+] mice after activation of *Gli1*[CreER/+] at 2 weeks of age. PIN prostatic intraepithelial neoplasia.

(Fig. 2a). A significant impairment of mPIN and adenocarcinoma development was observed in *Hi-Myc:Ar*[L/Y]:*Gli1*[CreER/+] mice in comparison with *Hi-Myc:Gli1*[CreER/+] controls when they both received TM at postnatal day 14, P14 (Fig. 2b). Grossly, prostates of 2- and 6-month-old *Hi-Myc:Ar*[L/Y]:*Gli1*[CreER/+] mice were significantly smaller and weighed less (*p* = 3.97 × 10[−8] and *p* = 0.003, respectively; Fig. 2c) than those of age-matched *Hi-Myc:Gli1*[CreER/+] littermates (Figs. 2d and 2e vs 2f and 2g). Pathological analyses showed mPIN3 and 4 as well as prostatic adenocarcinoma lesions in 2- and 6-month-old *Hi-Myc:Gli1*[CreER/+] mice, respectively, similar to the lesions reported in original Hi-Myc mice[18] (Fig. 2d, e, h). In contrast, only prostatic hyperplasia and PIN1 or 2 lesions revealed in prostate tissues of age-matched *Hi-Myc:Ar*[L/Y]:*Gli1*[CreER/+] counterparts (Fig. 2f–h). IHC analyses showed specific nuclear Myc staining in atypical and tumor cells in prostate tissues of both 2- and 6-month-old HiMyc mice (red arrows, Fig. 2d, e), demonstrating Myc-induced oncogenesis. Positive AR staining appeared in both epithelial and stromal cells in the above samples (pink and blue arrows,

Fig. 2d, e). In contrast, in samples of age-matched *Hi-Myc:Ar*[L/Y]:*Gli1*[CreER/+] mice, few atypical cells showed positive Myc, and AR staining appeared in epithelial cells but not in stromal cells (*p* < 0.01; pink arrows, Fig. 2f, g; Supplementary Fig. 1d). Additionally, less Ki67+ cells appeared in *Hi-Myc:Ar*[L/Y]:*Gli1*[CreER/+] mice than in *Hi-Myc:Gli1*[CreER/+] counterparts (*p* < 0.01; Fig. 2d, e vs f, g; Supplementary Fig. 1e). In this study, we also examined stromal AR action in Gli1-lineage cells to promote PIN and tumor progression by injecting TM to 2-month-old *Hi-Myc:Ar*[L/Y]:*Gli1*[CreER/+] and *Hi-Myc:Gli1*[CreER/+] mice and analyzed them at 6-months of age (Supplementary Fig. 1f). Mild pathological changes reflecting impaired PIN and prostate tumor development revealed in *Hi-Myc:Ar*[L/Y]:*Gli1*[CreER/+] mice in comparison with the age-matched *Hi-Myc:Gli1*[CreER/+] counterparts (Supplementary Fig. 1g–j). Taken together, these results demonstrate a promotional role of stromal AR signaling in Gli1-lineage cells to support prostatic epithelial oncogenesis and tumor development, which also provides experimental evidence to use these relevant models for in-depth molecular analyses.

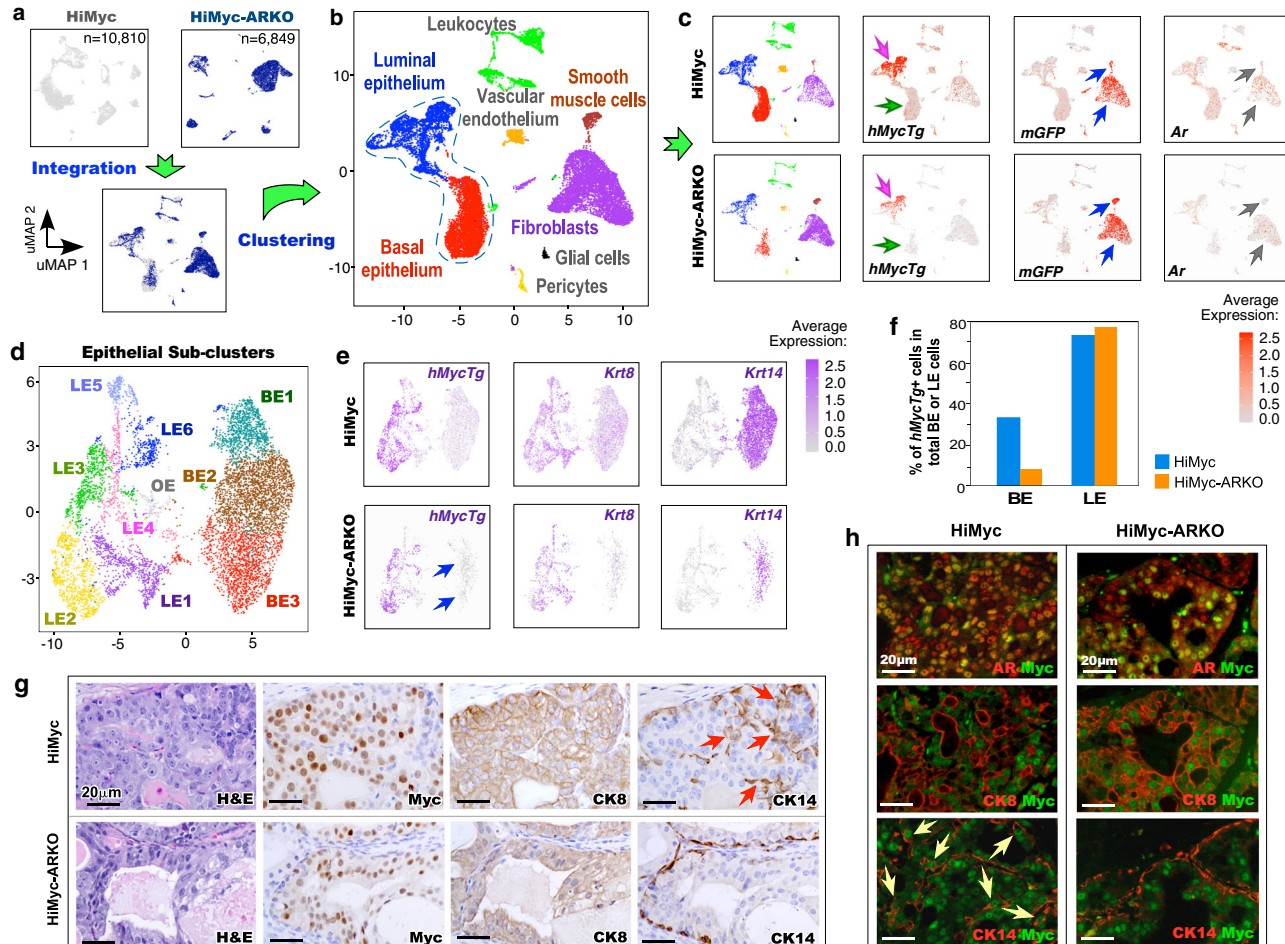

**Fig. 3 | Loss of AR in Gli1-lineage stromal cells impacts the basal epithelial compartment at a single cell resolution. a** Uniform Manifold Approximation and Projection (UMAP) visualization of 10,810 individual cells from *Hi-Myc:R26^{mTmG/+}:Gli1^{CreER/+}* prostates (top left, gray), 6849 individual cells from *Hi-Myc:R26^{mTmG/+}:Ar^{L/Y}:Gli1^{CreER/+}* prostates (top right, dark blue), and integrated cells (bottom). **b** UMAP representation showing cell-type cluster identities based on gene expression profiles. The dotted line delineates the prostatic epithelial cells. **c** UMAP plots separated by indicated genotype from **b** (Far left) and gene expression levels in *Hi-Myc:R26^{mTmG/+}:Gli1^{CreER/+}* (top) and *Hi-Myc:R26^{mTmG/+}:Ar^{L/Y}:Gli1^{CreER/+}* (bottom) mice. Color intensity indicates the scaled expression level in each cell. Pink and green arrows indicate luminal and basal epithelial cell clusters, respectively, and blue and grey arrows indicate stromal fibroblast and smooth muscle cell clusters, respectively.

**d** UMAP visualization of epithelial cells sub-clustered, re-clustered, and labeled by cell type. BE, basal epithelial cells; LE, luminal epithelial cells; OE, other epithelial cells. **e** Gene expression levels of markers for human *Myc* transgene (*hMycTg*), LE (*Krt8*), and BE (*Krt14*). **f** Graph showing the percentage of *hMycTg*⁺ cells per total basal or luminal cells in each genotype. Source data are provided as a Source Data file. **g** Representative images of H&E and IHC staining for Myc, cytokeratin 8 (CK8), and cytokeratin 14 (CK14) in adjacent prostate tissue sections of 3-month-old *Hi-Myc:R26^{mTmG/+}:Gli1^{CreER/+}* (top) and *Hi-Myc:R26^{mTmG/+}:Ar^{L/Y}:Gli1^{CreER/+}* (bottom) mice. Scale bars, 20 μm. Red arrows indicate CK14⁺ cells infiltrating towards the lumen. **h** Colocalization of Myc with either AR, CK8, or CK14 from indicated prostate tissues. Yellow arrows indicate Myc⁺CK14⁺ cells. Scale bars, 20 μm.

## Deletion of AR in stromal Gli1-lineage cells impairs prostatic basal epithelial cell-mediated oncogenesis

We developed *Hi-Myc:R26^{mTmG/+}:Gli1^{CreER/+}* and *Hi-Myc:R26^{mTmG/+}:Ar^{L/Y}:Gli1^{CreER/+}* mice, further referred to as HiMyc and HiMyc-ARKO mice later, respectively (Supplementary Fig. 2a, b), which showed similar pathological lesions as observed in *Hi-Myc:Ar^{L/Y}:Gli1^{CreER/+}* and *Hi-Myc:Gli1^{CreER/+}* counterparts but express *CreER* activated membrane green fluorescent protein (mGFP) in Gli1-lineage cells[19]. Given HGPIN lesions and prostatic adenocarcinoma lesions developed in 2- and 6-month-old *Hi-Myc:Gli1^{CreER/+}* mice, we performed single-cell RNA sequencing (scRNA-seq) using prostate tissues of 3-month-old HiMyc and HiMyc-ARKO mice to assess the molecular mechanisms for stromal AR signaling in early prostate epithelial tumorigenesis. After the data processing (Supplementary Fig. 2c–j; also see "Methods"), both HiMyc and HiMyc-ARKO samples were visualized individually using the uniform manifold approximation and projection (UMAP) and then merged and clustered for analysis of molecular changes induced by stromal AR deletion Gli1-lineage cells[20] (Fig. 3a). A total of 8 cell subsets were

identified based on their transcriptomic profiles[21–24] (Fig. 3b; Supplementary Fig. 2k–l) and aligned in separated UMAP plots of HiMyc and HiMyc-ARKO samples (Fig. 3c), demonstrating their comparable cellular properties. UMAP expression plots showed that *hMycTg* expression was comparable in luminal epithelial cell sets of both samples but reduced in basal epithelial cell sets in ARKO samples in comparison with HiMyc samples (pink and green arrows, Fig. 3c). *Gli1-CreER* activated *mGFP* expression was observed in fibroblasts and smooth muscle cell subsets of both samples, further confirming the stromal cell properties of Gli1-lineage cells (blue arrows, Fig. 3c). Reduction of *Ar* expression appeared in stromal cell subsets of HiMyc-ARKO mice (grey arrows, Fig. 3c). To gain higher resolution of *hMycTg* expression induced cellular and molecular changes, we separated epithelial cells from other non-epithelial cells, and re-clustered them. Ten epithelial cell clusters were identified, including 3 basal, 6 luminal, and 1 other epithelial cluster, OE (Fig. 3d; Supplementary Fig. 3a, b). The expression of *hMycTg* appeared comparable in luminal epithelial cells with high *Krt8* expression in both HiMyc and HiMyc-ARKO samples, but

reduced in *Krt14*+ basal epithelial cells in HiMyc-ARKO samples (arrows, Fig. 3e). Specifically, the percentage of *hMycTg*+ basal cells was significantly higher in HiMyc than in HiMyc-ARKO mice ($p < 0.01$; Fig. 3f). IHC analyses showed Myc+ cells were clearly overlaid with CK8+ cells in PIN lesions on adjacent prostate tissues sections of both HiMyc and HiMyc-ARKO mice (top panel, Fig. 3g). However, noticeable overlays between Myc+ and CK14+ cells only appeared in PIN lesions of HiMyc mice (red arrows, Fig. 3g). In contrast, CK14+ cells were mainly localized in basal cell layers in HiMyc-ARKO samples. Co-immunofluorescent analyses (Co-IF) further identified a significant increase of CK14+ Myc+ atypical cells in PIN lesions of HiMyc samples ($p < 0.01$; yellow arrows, Fig. 3h; Supplementary Fig. 3c). Myc+ luminal cells overlaying with AR and CK8 staining also appeared in both HiMyc and HiMyc-ARKO samples (Fig. 3h). Using triple-IF approaches, we further identified CK8+ CK14+ and CK8+ CK14+ Myc+ cells in HiMyc samples, indicating the intermediate cell properties of some CK14+ cells in the samples (white arrows, Supplementary Fig. 3d). Prostatic basal epithelial cells are directly adjacent to stromal cells and possess progenitor properties[23,25]. Specifically, Myc+ basal epithelial cells have been shown to function as prostate tumor-initiating cells in Hi-Myc mice[26]. Therefore, our data of identifying reduced atypical Myc+ basal cells in PIN tissues of HiMyc-ARKO mice implicate an underlying mechanism by which stromal AR deletion in Gli1-lineage cells impairs prostatic oncogenesis and tumor development.

### Deletion of AR in stromal Gli1-lineage cells impedes IGF1-induced Wnt/β-catenin activation in prostatic basal epithelia

To assess stromal AR-induced molecular changes in prostatic basal epithelium, we performed gene set enrichment analysis (GSEA) using the differentially expressed genes (DEGs) identified in *hMycTg*+ basal cells between HiMyc and HiMyc-ARKO samples (Supplementary Data 1). Significantly enriched IGF1 receptor (IGF1R), Wnt, and other signaling pathways directly related to prostate tumorigenesis were identified in *hMycTg*+ basal epithelial cells of HiMyc samples (Fig. 4a; Supplementary Fig. 3e). Emerging evidence has shown a prominent role of IGF1/IGF1R signaling activation in inducing prostate cancer development and progression in both mouse models and clinical studies[27]. IGF1R activation can induce Wnt/β-catenin signaling to promote prostate tumorigenesis[28–30]. Importantly, IGF1 and Wnt/β-catenin signaling activation directly contributes to androgen-induced prostate oncogenesis and tumor development[31]. Given these lines of relevant scientific evidence, we examined transcriptomic changes and identified increased expression of *Igf1r* and its downstream genes, *Hras, Irs2*, and *Akt1* ($p < 0.01$; top panel, Fig. 4b), as well as Wnt signaling effectors and targets, *Ctnnb1, Ccnd1, Cd44*, and *Tcf7l2* in *hMycTg*+ basal epithelial cells of HiMyc samples in comparison to HiMyc-ARKO samples ($p < 0.01$; bottom panel, Fig. 4b; Supplementary Fig. 3f, g). A significant positive correlation between the expression of *Igf1r* and its downstream target genes, *Hras, Irs2*, and *Akt1* as well as Wnt signaling effectors, *Ctnnb1*, and targets, *Ccnd1, Cd44*, and *Tcf7l2* (Spearman $r = 0.3, 0.2, 0.3, 0.4, 0.2, 0.3$, and $0.3$, respectively; $p = 1.44 \times 10^{-29}$, $6.27 \times 10^{-18}$, $7.15 \times 10^{-35}$, $1.42 \times 10^{-63}$, $2.93 \times 10^{-5}$, $3.12 \times 10^{-23}$, and $5.14 \times 10^{-42}$, respectively) was identified in *hMycTg*+ basal epithelial cells of HiMyc mice but not in the counterparts of HiMyc-ARKO mice (Fig. 4c), demonstrating the specific activation of IGF1R and Wnt signaling pathways induced by stromal AR in prostatic basal cells. Real-time quantitative reverse transcription PCR (qRT-PCR) analyses further identified upregulated expression of *hMycTg, Igf1r*, and its downstream genes, *Hras, Grb2, Irs2*, and *Akt1*, as well as *Ctnnb1* and its downstream targets, *Cd44, Tcf7l2*, and *Ccnd1* in sorted prostatic basal epithelial cells of HiMyc versus those of HiMyc-ARKO mice ($p < 0.05$; Fig. 4d). It has been shown that aberrant activation of IGF1 signaling can either directly or via activating AKT and GSK3β axes stabilize cellular β-catenin to augment its activity for prostate tumor

growth[28,29,32]. We, therefore, assessed the effect of IGF1 signaling in activating Wnt/β-catenin axes in prostatic atypical basal cells. IHC analyses further detected positive staining for IGF1R, phosphorylated IGF1R (pIGF1R), and CyclinD1, a downstream target of β-catenin, in atypical cells of PIN lesions in HiMyc mouse tissues (Supplementary Fig. 3h1–4 vs 3i1–4), suggesting activated IGF1R and Wnt signaling pathways. Using triple-IF approaches, we further identified co-expression of Myc and CK14, as well as pIGF1R, phosphorylated AKT, phosphorylated GSK3β, cytoplasmic and nuclear β-catenin, CyclinD1, or TCF7L2 in atypical basal cells in PIN lesions of HiMyc mice but not those of HiMyc-ARKO mice ($p < 0.01$; yellow arrows, top panel, Fig. 4e; Supplementary Fig. 3j). The above regulatory loop was further validated with co-IF analyses in HGPIN tissues of HiMyc mice (Supplementary Fig. 3k). These data demonstrate the impairment of IGF1R and canonical Wnt signaling pathways in prostatic basal cells of HiMyc-ARKO mice, providing mechanistic insight into stromal AR in Gli1-lineage cells to regulate prostatic epithelial tumorigenesis.

### Deletion of AR expression induces robust IGFBP3 expression in stromal Gli1-lineage cells

To gain in-depth insight into prostatic Gli1-lineage cells as stromal niches, we separated stromal cells from other types of cells and re-clustered them into five fibroblast (FB) and smooth muscle cell clusters (Fig. 5a, b; Supplementary Fig. 4a–b). Reduced *Ar* expression was detected in stromal cell clusters of HiMyc-ARKO mice while comparable *Gli1-CreER* driven *mGFP* expression appeared in stromal cells of both samples, demonstrating selective deletion of *Ar* expression in Gli1-lineage stromal cells (Fig. 5c–d). We then analyzed the DEGs of *mGFP*+ FB between HiMyc and HiMyc-ARKO samples to explore their role as cancer-associated fibroblasts (CAF)[33] (Supplementary Data 2). Intriguingly, we identified the *Igfbp3, insulin-like growth factor-binding protein 3*, as one of the most highly expressed genes in FB of HiMyc-ARKO (the bottom panel, Fig. 5d, e). IGFBP3 is a key regulator of IGF1 pathways and can bind IGF1/2 to block their access to their receptors for activation[34]. Increased *Igfbp3* expression was further identified in *mGFP*+ FB of HiMyc-ARKO samples (Fig. 5f). Intriguingly, an inverse correlation between *Ar* and *Igfbp3* expression was identified in *mGFP*+ FB (middle and bottom panels, Fig. 5d; Supplementary Fig. 4c, d). A significant increase in *Igfbp3* alongside reduced *Ar* expression was further confirmed in sorted mGFP+ cells from HiMyc-ARKO prostate tissues in comparison to those from HiMyc counterparts using qRT-PCR analyses ($p = 1.09 \times 10^{-6}$ and $p = 3.36 \times 10^{-6}$, respectively; Fig. 5g). In alignment with the above results, co-IF analyses showed the significant co-occurrence of reduced AR and increased IGFBP3 expression in mGFP+ and VIM+ fibroblasts of HiMyc-ARKO prostate tissues ($p = 3.35 \times 10^{-6}$ and $p = 1.09 \times 10^{-6}$, respectively; left vs. right panel, Fig. 5h; Supplementary Fig. 4e). These data demonstrate an inverse relationship between AR and IGFBP3 expression in prostatic stromal Gli1-lineage cells, implicating a regulatory mechanism by which stromal AR regulates IGF1 signaling to facilitate prostatic epithelial oncogenesis and tumor development.

### Stromal AR in Gli1-lineage cells represses Sp1-mediated Igfbp3 expression

Regulation of *Igfbp3* transcription is mediated mainly through Sp1 transcription factor[35]. Previous studies have demonstrated that the AR can interfere Sp1 binding to the target promoters and repress Sp1-mediated transcription[36,37]. To examine the regulatory role of AR on IGFBP3 expression, we performed chromatin immunoprecipitation (ChIP) assays to assess the direct involvement of AR on Sp1-mediated IGFBP3 transcription. We observed increased recruitment of Sp1 within the *Igfbp3* promoter regions ($p = 0.002$) in sorted AR-deficient mGFP+ cells from HiMyc-ARKO mice in comparison to the samples from HiMyc mice, but no difference in the control *Untr4* locus[38] (Fig. 5i). Accordingly, less AR occupancy on the *Igfbp3* promoter

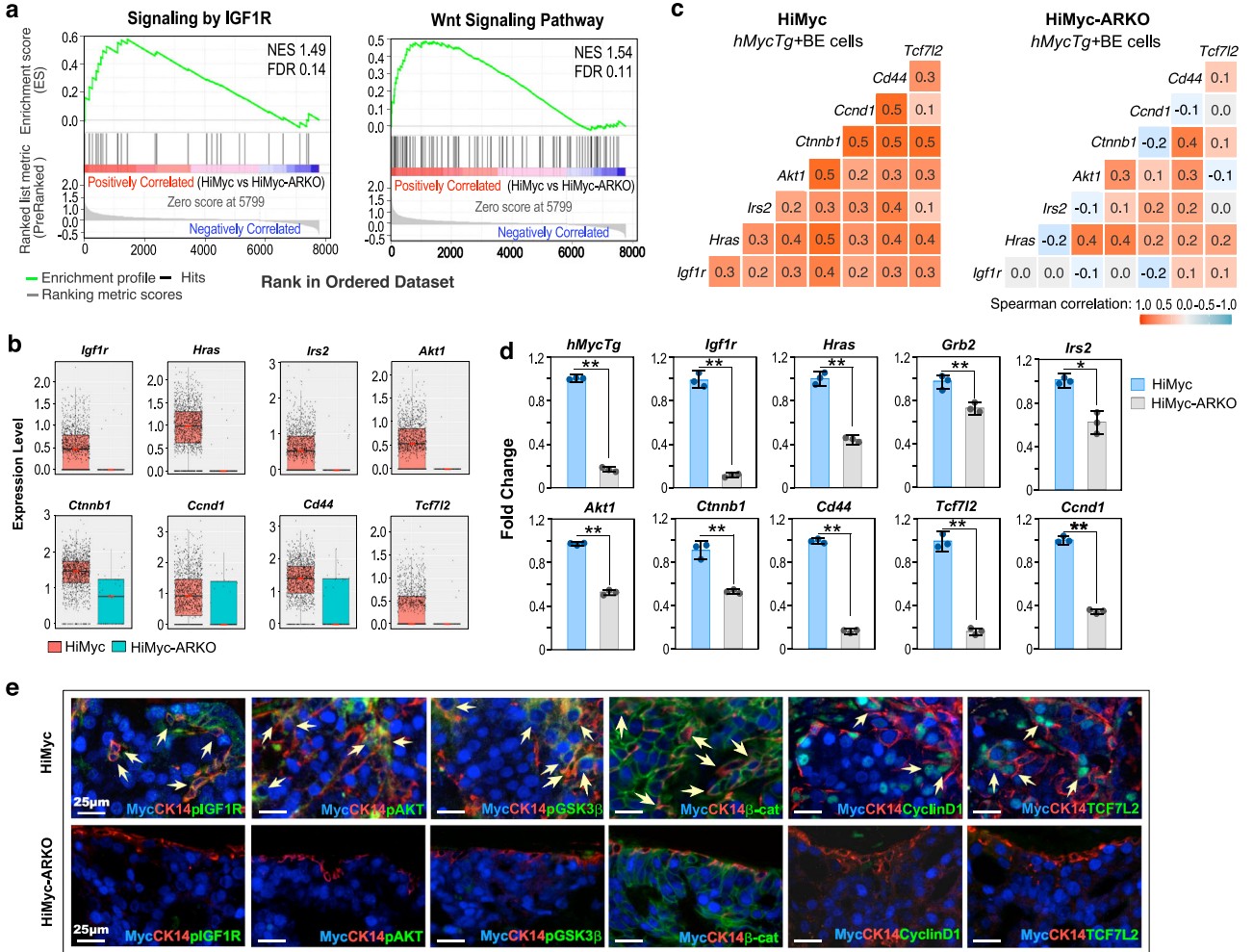

**Fig. 4 | Loss of AR in Gli1-lineage cells reduces IGF and Wnt signaling in basal epithelial cells. a** Gene Set Enrichment Analysis (GSEA) enrichment plots depicting insulin-like growth factor 1 receptor (IGF1R) and Wnt signaling pathways significantly up-regulated in $hMycTg^+$ basal epithelial cells in $Hi-Myc:R26^{mTmG/+}:Gli1^{CreER/+}$ versus $Hi-Myc:R26^{mTmG/+}:Ar^{L/Y}:Gli1^{CreER/+}$. See also "Methods" section. NES, normalized enrichment score; FDR, false discovery rate. **b** Box plots representing scaled expression data for IGF1R (top) or Wnt (bottom) downstream genes in $hMycTg^+$ basal cells from indicated genotypes (HiMyc, $n = 1363$; HiMyc-ARKO, $n = 36$). Pink lines mark the median; top and bottom lines of boxes indicate the boundaries of the first and third quartiles, respectively; the top and bottom whiskers show the maximum and minimum values, respectively, excluding outliers. **c** Heatmap of

pairwise Spearman correlation between the indicated gene expression in $hMycTg^+$ basal cells from indicated genotypes. Colors reflect the level of the correlation coefficient. Numbers show the correlation coefficient. **d** qPCR analysis of gene expression as indicated in the figure shown as fold change in sort-purified basal cells from indicated genotypes. Data are represented as mean ± SD of three biological replicates. Two-sided Student's $t$ test, $^*p < 0.05$, $^{**}p < 0.01$. The exact $p$ values and related source data are provided as a Source Data file. **e** Representative images of triple-immunofluorescent (IF) staining for indicated antibodies on prostate tissue sections from $Hi-Myc:R26^{mTmG/+}:Gli1^{CreER/+}$ (top) and $Hi-Myc:R26^{mTmG/+}:Ar^{L/Y}:Gli1^{CreER/+}$ (bottom). Yellow arrows indicate triple-positive stained cells. Scale bars, 25 μm.

regions revealed in samples of HiMyc-ARKO mice compared to those of HiMyc and wild-type control mice ($p = 0.011$ and $p = 0.003$, respectively; Supplementary Fig. 4f). These data implicate that AR deletion attenuates the repression of Sp1 recruitment on the *Igfbp3* promoter in Gli1-lineage FB, augmenting *Igfbp3* transcription. Increased IGFBP3 expression was also identified in prostatic stromal cells of HiMyc-ARKO mice when the *Ar* was deleted at 2 months of age (Supplementary Fig. 4g1–i3), corresponding to the impaired PIN and prostate tumor development in these mice (Supplementary Fig. 1f–j). Taken together, these data uncover a regulatory mechanism by which AR deletion induces IGFBP3 expression in Gli1-lineage FB to block IGF1-signaling activation, further hindering prostatic basal epithelial cell-mediated oncogenesis and tumor development.

### Stromal AR in Gli1-lineage FB converts the cellular properties of CAF

To assess the role of stromal AR in Gli1-lineage FBs, we analyzed DEGs between FBs of HiMyc and HiMyc-ARKO samples and identified

expression of CAF cellular markers[33], including *Pdgfrβ, Twist1, Sox9, Foxf1, Il11*, and *Cxcl10* (Supplementary Fig. 4j). Using qRT-PCR analyses, we further demonstrated reduced expression of those CAF markers in sorted mGFP+ cells from HiMyc-ARKO samples in comparison to those of HiMyc counterparts ($p < 0.001$; Fig. 5j). Triple-IF analyses also showed the specific co-localization between PDGFRβ or Twist1 with AR and *Gli1-CreER* activated mGFP expression in PIN tissues of HiMyc mice (blue arrows in the top panel, Fig. 5k; Supplementary Fig. 4k1–4), but not in ones of HiMyc-ARKO mice ($p < 0.01$; bottom panel, Fig. 5k; Supplementary Fig. 4l1–4m). Significantly up-regulated expression of CAF cellular markers, including *Pdgfrβ* ($p = 9.13 \times 10^{-7}$), *Twist1* ($p = 2.05 \times 10^{-32}$), *Sox9* ($p = 2.05 \times 10^{-32}$), *Foxf1* ($p = 8.40 \times 10^{-37}$), *Il11* ($p = 3.24 \times 10^{-36}$), and *Cxcl10* ($p = 3.38 \times 10^{-50}$) was further identified in the FB1 cell cluster of HiMyc samples (Fig. 5l; Supplementary Fig. 4n, o). Analyses of transcriptomic changes in FB1 cluster between *mGFP+ Ar+* FB of HiMyc and *mGFP+ Ar-* FB of HiMyc-ARKO showed up-regulation of *Ar*, AR downstream target genes, and Wnt ligands, and down-regulation of P53 and Sp1 downstream targets as well as IGFBP3

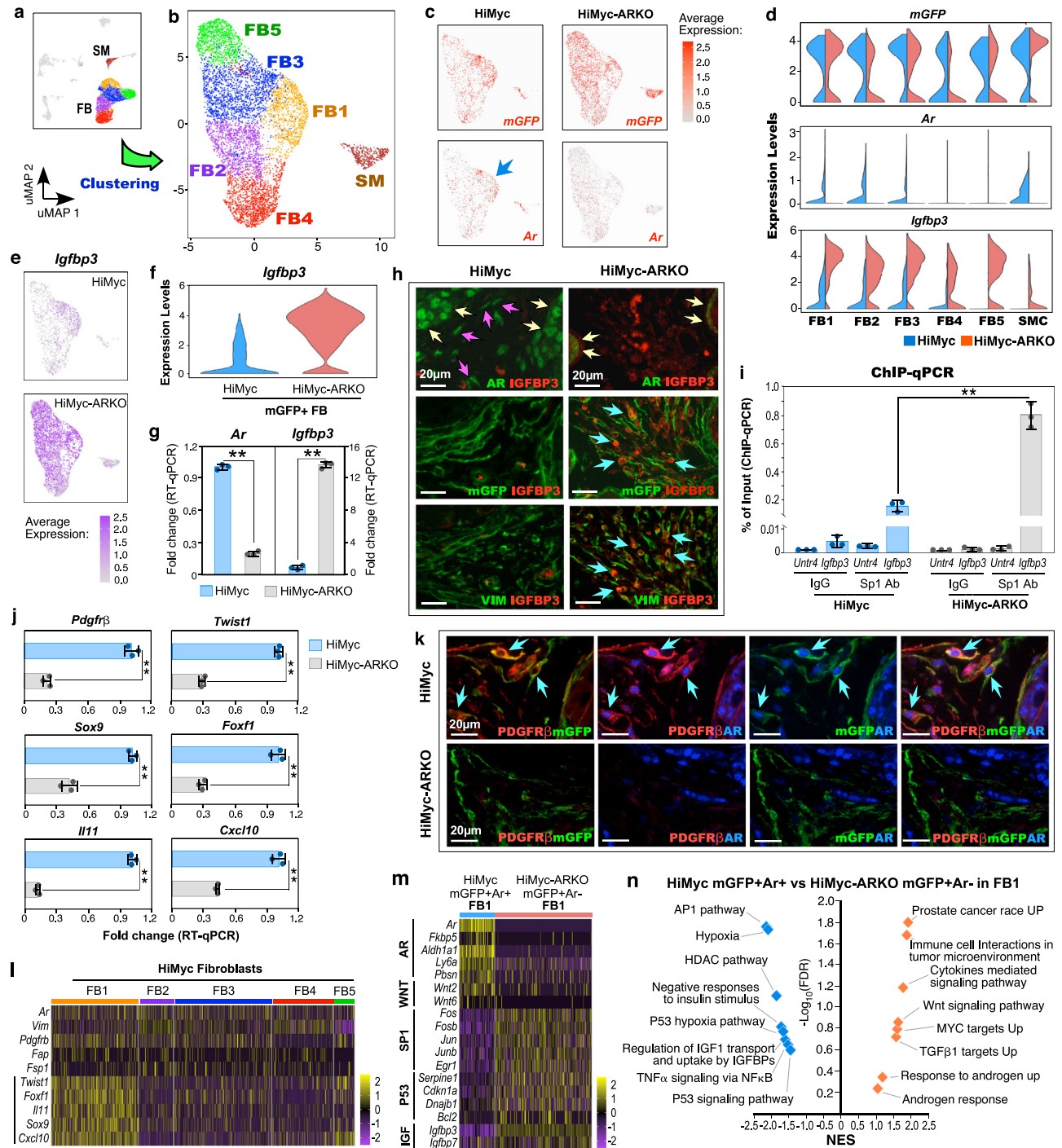

**Fig. 5 | Identifying AR loss-induced molecular changes in Gli1-lineage stromal cells. a–b** UMAP plots of stromal fibroblast (FB) and smooth muscle (SM) cell clusters from *Hi-Myc:R26^{mTmG/+}:Gli1^{CreER/+}* and *Hi-Myc:R26^{mTmG/+}:Ar^{L/Y}:Gli1^{CreER/+}*. **c** Gene expression levels of *mGFP* and *Ar* in UMAP plots separated by indicated genotypes. Color intensity indicates the scaled expression level in each cell. Blue arrow indicates *Ar*-expressing cells from FB1 in *Hi-Myc:R26^{mTmG/+}:Gli1^{CreER/+}*. **d** Violin plots visualizing the expression levels of *mGFP*, *Ar*, and *insulin-like growth factor binding protein 3 (Igfbp3)* in stromal clusters of each genotype. **e** Gene expression levels of *Igfbp3* in UMAP plots of stromal FB and SM cells. Color intensity indicates the scaled expression level in each cell. **f** Violin plots of *mGFP^+* FB cells. **g** qPCR analysis of *Ar* and *Igfbp3* expression shown as fold change in FACs-sorted mGFP^+ cells from prostate tissues of indicated genotypes. **h** Co-localization of IGFBP3 with either AR, mGFP, or Vimentin (Vim) from indicated prostate tissues. Yellow and pink arrows indicate AR^+ epithelial and stromal cell, respectively. Blue arrows indicate IGFBP3^+ stromal cells co-expressing with either mGFP or Vim. Scale bars, 20 μm. **i** Sp1

Chromatin immunoprecipitation qPCR analysis of the *Igfbp3* promoter regions, and negative control (*Untr4*) shown as percent input. **j** qPCR analysis of cancer-associated fibroblast (CAF) markers shown as fold change in sort-purified mGFP^+ cells. **k** Representative images of triple-IF staining for indicated antibodies on prostate tissue sections from indicated genotypes. Blue arrows indicate cells showing double- or triple-positive with the indicated proteins. Scale bars, 20 μm. **l** Heatmap displaying gene expression of *Ar, Vim*, and CAF markers across FB clusters of *Hi-Myc:R26^{mTmG/+}:Gli1^{CreER/+}*. **m** Heatmap depicting expression of genes associated with indicated pathways. **n** GSEA showing the enrichment of indicated gene signatures in *mGFP^+Ar^+* FB1 cells from *Hi-Myc:R26^{mTmG/+}:Gli1^{CreER/+}* versus *mGFP^+Ar^-* FB1 cells from *Hi-Myc:R26^{mTmG/+}:Ar^{L/Y}:Gli1^{CreER/+}*. See also "Methods" section. NES, normalized enrichment score; FDR, false discovery rate. In **g, i**, and **j**, data are represented as mean ± SD of three biological replicates. Two-sided Student's *t* test, \*\**p* < 0.01. The exact *p* values and related source data are provided in the "Source Data file".

and 7 (Fig. 5m; Supplementary Data 3). GSEA based on the above DEGs identified the enrichment in upregulated pathways that promote prostate cancer ($p = 0.006$) and androgen signaling activation ($p = 0.016$), and downregulated pathways related to cell hypoxia ($p = 0.019$), negative responses to insulin and IGF1 ($p = 0.015$), and P53-mediated signaling ($p = 0.025$) in the *mGFP+ Ar+* FB1 of HiMyc samples (Fig. 5n). These data provide in-depth mechanistic insight into the regulatory role of stromal AR in Gli1-lineage FB acting as CAF to support prostate epithelial tumor cell growth and progression.

### IGFBP3 represses IGF1/IGF1R signaling-induced prostate epithelial cell growth

Identifying the regulation of stromal AR in IGFBP3 expression in Gli1-lineage FB is intriguing, suggesting an underlying mechanism for androgen-mediated IGF1 signaling activation in prostate oncogenesis. To directly assess the effect of IGF1 on prostatic basal epithelium, we developed organoid cultures using enriched basal epithelial cells through cell sorting from microscopically confirmed PIN tissues of HiMyc and HiMyc-ARKO mice (see the "Methods"[39,40]; Supplementary Fig. 5a–c). Organoids derived from HiMyc-ARKO mice showed retarded growth in comparison with those from HiMyc mice at earlier culture time points ($p = 7.36 \times 10^{-5}$, $p = 0.004$, and $p = 0.25$, respectively; Supplementary Fig. 5d). However, the growth difference gradually diminished at day 6, implicating the effects of microenvironment on their growth. To mimic the niche effects as observed in our in vivo models, we treated epithelial organoids with IGF1, IGF1 + IGFBP3, or vehicle. Measuring organoid forming efficiency and average sizes of individual organoids showed significant increases in IGF1-treated samples compared to IGF1 + IGFBP3-treated counterparts in both HiMyc and HiMyc-ARKO mice ($p < 0.001$), demonstrating the critical role of IGFBP3 in neutralizing IGF1 activity (Fig. 6a–c; Supplementary Fig. 5e1, 5f1). There is no significant difference between IGF1-treated epithelial organoids derived from HiMyc and HiMyc-ARKO samples (forming efficiency, $p = 0.590$ and organoid size, $p = 0.658$). Histologically, IGF1-treated groups in both genotypes showed severe pathological lesions featuring multilayer atypical cells growing into the lumen and as cribriform structures, whereas IGF1 + IGFBP3- or vehicle-treated groups displayed mild pathologic changes (bottom panel, Fig. 6a; Supplementary Fig. 5e2–3, 5f2–3). These results suggest epithelial organoids derived from atypical basal cells of HiMyc and HiMyc-ARKO samples, after growing in a new microenvironment, possessing comparable abilities in response to IGF1-induced cell growth. IHC analyses showed more Myc+ atypical cells in IGF1-treated organoids than in IGF1 + IGFBP3-treated or vehicle-treated counterparts in both genotype samples ($p < 0.001$; top panel, Fig. 6d; Supplementary Fig. 5g1, h1, i). In the IGF1-treated organoids, CK14+ cells appear to overlay with Myc+ cells and infiltrate in PIN lesion, in contrast to CK14+ cells only localized on the basal layers in the IGF1 + IGFBP3 treated organoids (middle panel, Fig. 6d). Both AR and CK8 positive cells were also observed in the above basal cell derived organoids (Fig. 6d), demonstrating the ability of CK14+ cells to differentiate into luminal cells. Positive staining for pIGF1R and cytoplasmic and nuclear staining of β-catenin revealed in atypical cells in the above IGF1-treated organoids from HiMyc and HiMyc-ARKO epithelia, but negative or reduced staining of those proteins appeared in IGF1 + IGFBP3- or vehicle-treated counterparts (bottom panels, Fig. 6d; Supplementary Fig. 5g3–4, 5h3–4). The above data of identifying increased cytoplasmic and nuclear β-catenin expression in IGF1-treated organoid samples further demonstrates the promotional role of Wnt/β-catenin activation in prostatic epithelial growth.

To directly assess the effect of Wnt/β-catenin in IGF1-induced prostatic basal epithelial growth, we treated organoids with Wnt inhibitors, ICG-001[41] and iCRT3[42] in combination with IGF1 (Fig. 6e, f; Supplementary Fig. 6a–c). It has been shown that iCRT3 can disrupt

the interaction between AR and β-catenin and inhibit AR-mediated transcription and cell growth in prostate cancer cells[42]. We observed more and larger organoids in IGF1-treated samples than those treated with IGF1 and Wnt inhibitors (Top panel, Fig. 6e, f). Measuring organoid forming efficiency and average size of individual organoids showed a significant decrease in samples treated with IGF1 and Wnt inhibitors, ICG-001 or iCRT3, in comparison to those treated with IGF1 only ($p < 0.001$; Supplementary Fig. 6a, b). Histological analyses also showed minor pathological changes in the former but typical HGPIN lesions in the latter (bottom panels, Fig. 6e, f). IHC further showed decreased expression of cytoplasmic and nuclear β-catenin, β-catenin downstream targets, Myc, Cyclin D1, and Ki67 in samples treated with Wnt inhibitors and IGF1 compared to those treated with IGF1 only (Supplementary Fig. 6c). Taken together, these data demonstrate the promotional role of Wnt/β-catenin activation in IGF1-induced prostate tumor cell growth.

### Stromal AR promotes prostatic basal epithelial oncogenesis through the IGFBP3-IGF1/IGF1R and Wnt/β-catenin regulatory loop

To directly examine the regulatory role of increased IGFBP3 on IGF1 in prostate stromal cells, we isolated mGFP+ stromal cells from both HiMyc and HiMyc-ARKO mice through cell sorting and measured the level of IGF1 in the stromal cell-conditioned medium. Higher levels of secreted IGF1 were detected in the stromal cell-conditioned medium prepared from HiMyc mice ($p = 1.79 \times 10^{-6}$; Fig. 7a). Prostate epithelial organoids treated with the stromal cell-conditioned medium prepared from HiMyc samples showed a significant increase in cell growth in comparison with those treated with the conditioned medium of HiMyc-ARKO stromal cells (Fig. 7b). Pre-incubation of IGFBP3 with the conditioned medium reduced its effects in inducing organoid cell growth when both organoid forming efficiency and average sizes of individual organoids were measured ($p < 0.001$; Supplementary Fig. 6d, e). IHC analyses showed positive staining for phosphorylated IGF1R, cytoplasmic and nuclear β-catenin, Myc, and Cyclin D1 in samples treated with the conditioned medium from HiMyc stromal cells but not in other samples (Supplementary Fig. 6f), directly demonstrating IGF1 activity in the above-conditioned medium. Specifically, positive staining for cytoplasmic and nuclear β-catenin and its downstream target, Cyclin D1 in the above samples further demonstrate the regulatory mechanisms for IGF1-induced Wnt/β-catenin activation in prostate epithelial cells. Altogether, these data provide additional evidence to demonstrate the role of stromal AR in Gli1-lineage cells, acting as a niche to support prostate epithelial cell growth through IGF1-induced Wnt signaling activation.

Using triple-IF analyses, we further assessed the reciprocal regulation between prostatic stromal IGFBP3 expression and epithelial IGF1R activation. Multilayered atypical Myc+CK14+ basal epithelial cells were observed adjacent to mGFP+ Gli1-lineage cells within PIN lesions of HiMyc samples in contrast to the single layer of Myc+ CK14+ basal epithelial cells in HiMyc-ARKO samples (yellow arrows, Fig. 7c), in which IGFBP3+ mGFP+ Gli1-lineage cells were mainly observed (blue arrows, Fig. 7c). Accordingly, positive staining of pIGF1R only appeared in HiMyc but not in HiMyc-ARKO samples (white arrows, Fig. 7c). Increased Ki67+ staining also appeared in prostatic basal epithelium in HiMyc versus HiMyc-ARKO samples, indicating the promotional role of IGF1R activation in prostatic tumor epithelia ($p = 5.66 \times 10^{-5}$; Fig. 7c; Supplementary Fig. 6g). These similar results were also observed in prostate tissues of 6-month-old HiMyc and HiMyc-ARKO mice when they received TM at 2 months of age (Supplementary Fig. 6h), validating the regulatory role of IGFBP3 on IGF1R1 signaling. Taken together, these data in combination with other results elucidate that AR loss attenuates the repression on Sp1-mediated IGFBP3 transcription resulting in elevated IGFBP3 in Gli1-lineage FB, which reduces IGF1 binding to IGF1R

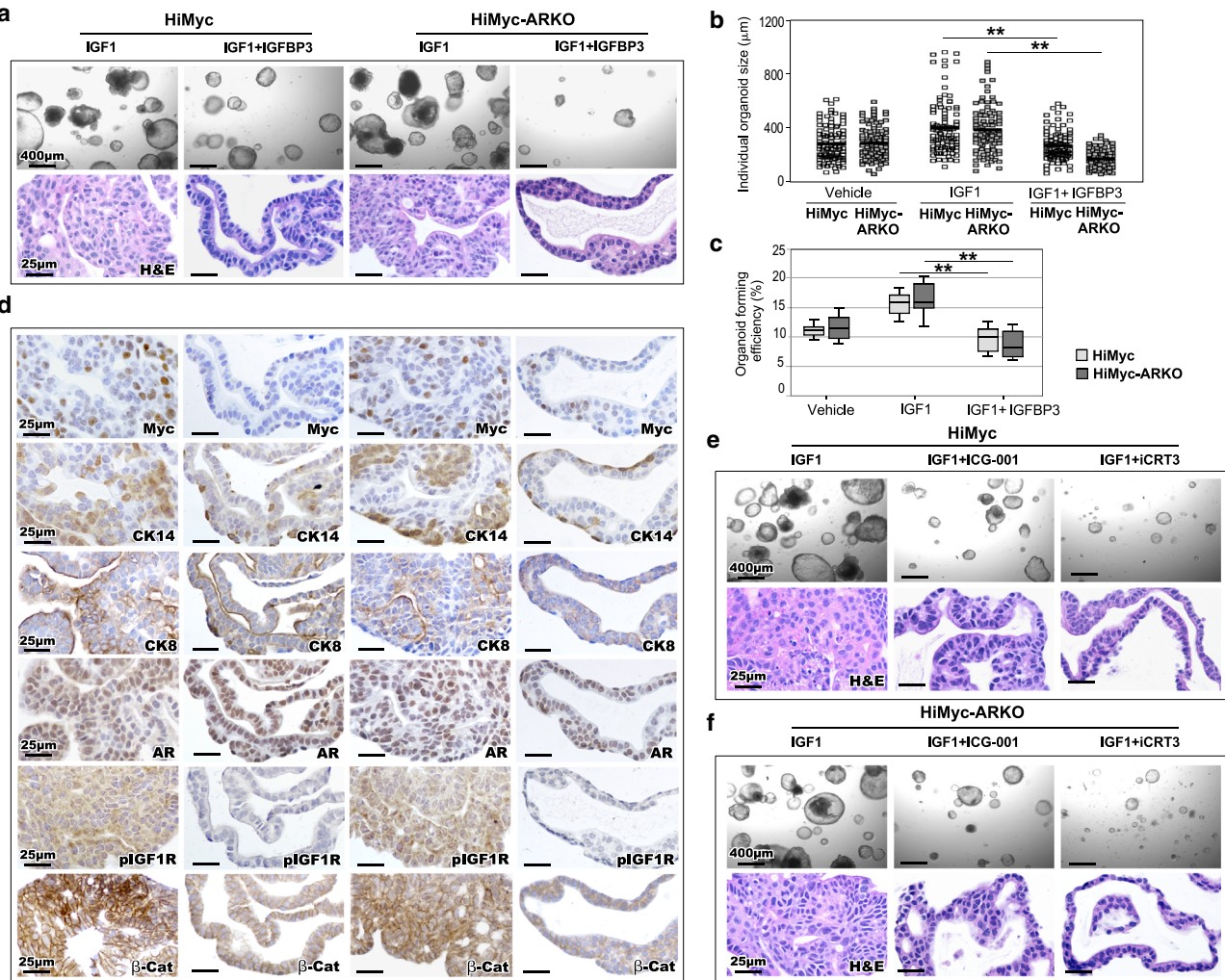

**Fig. 6 | Identifying the reciprocal effect of stromal AR action on prostatic epithelial organoid oncogenic transformation and growth. a** Organoid culture of prostatic epithelial cells from 12-week-old *Hi-Myc:R26^mTmG/+:Gli1^CreER/+* and *Hi-Myc:R26^mTmG/+:Ar^L/Y:Gli1^CreER/+* mice, following treatment with vehicle, IGF1 (100 ng/ml), or IGF1 (100 ng/ml) + IGFBP3 (1 µg/ml). Representative of brightfield images and HE staining showing organoid morphology and structure observed in indicated groups. Scale bars, 400 µm and 25 µm. **b** Quantification of individual organoid size. Organoids (*n* = 100) per group examined over three independent experiments. The center line represents the median value in each group. **c** Quantification of organoid forming efficiency showing the percentage of organoids above 50 µm diameter per total cells seeded at day 0 in a well. The center line represents the median value, the box borders represent the lower and upper quartiles (25% and 75% percentiles,

respectively) and the ends of the bottom and top whiskers represent the minimum and maximum values, respectively, for eight independent samples over three biological replicates. **d** Representative images of IHC staining with the indicated antibodies. Scale bars, 25 µm. **e–f** Organoid culture of prostatic epithelial cells from 12-week-old *Hi-Myc:R26^mTmG/+:Gli1^CreER/+* and *Hi-Myc:R26^mTmG/+:Ar^L/Y:Gli1^CreER/+* mice, following treatment with IGF1 (100 ng/ml), IGF1 (100 ng/ml) + ICG-001 (5 µM), or IGF1 (100 ng/ml) + iCRT3 (5 µM). Representative brightfield images and HE staining showing organoid morphology and structure observed in indicated groups. Scale bars, 400 µm and 25 µm. **b, c** Two-sided Student's *t*-test for IGF1-treated groups versus IGF1 + IGFBP3-treated groups in HiMYC or HiMYC-ARKO, **p* < 0.01. Source data and the exact *p*-values are provided in the "Source Data file".

on adjacent prostatic basal epithelial cells and diminishes IGF1R-mediated Wnt/β-catenin activation (Fig. 7d).

Emerging evidence has shown the prostatic stem/progenitor cell properties of human prostatic basal epithelial cells[1]. Human prostatic cancer cells with basal cell gene signatures are also linked to advanced, metastatic, and castration-resistance tumor phenotypes[43]. To explore the clinical relevance of stromal AR action as a tumor niche, we compared GSEA based on DEGs of basal versus luminal cell in human prostate tumors with those in our mouse models, and identified similar enriched signaling pathways between human samples and HiMyc mice. Specifically, both IGF1 and canonical Wnt signaling pathways were significantly enriched (Fig. 7e and Supplementary Data 4). These data provide additional and clinically relevant evidence demonstrating the critical role of reciprocal interactions between stromal AR signaling in Gli1-lineage cells and prostatic basal epithelial tumorigenesis.

## Discussion

Prostate tumorigenesis is solely governed through AR-mediated signaling pathways[44,45]. However, most previous studies were mainly focused on prostate tumor cell-intrinsic mechanisms[4–6]. The roles and underlying mechanisms for stromal AR action in promoting prostate epithelial oncogenesis are largely unknown. Additionally, the cellular identity of prostate stromal cells that act as tumor niches is also unclear. Data from this study provide multiple lines of scientific evidence demonstrating a critical role of AR signaling in stromal Gli1-lineage cells in supporting prostate epithelial cell tumorigenesis, leading to the development of different therapeutic strategies for treating advanced prostate cancer.

In this study, we identified a critical role of AR in stromal Gli1-lineage cells to regulate IGF1 signaling pathways through IGFBP3. Using scRNA-seq and other experimental analyses, we observed a robust increase in *Igfbp3* expression in AR-deficient stromal Gli1-lineage cells,

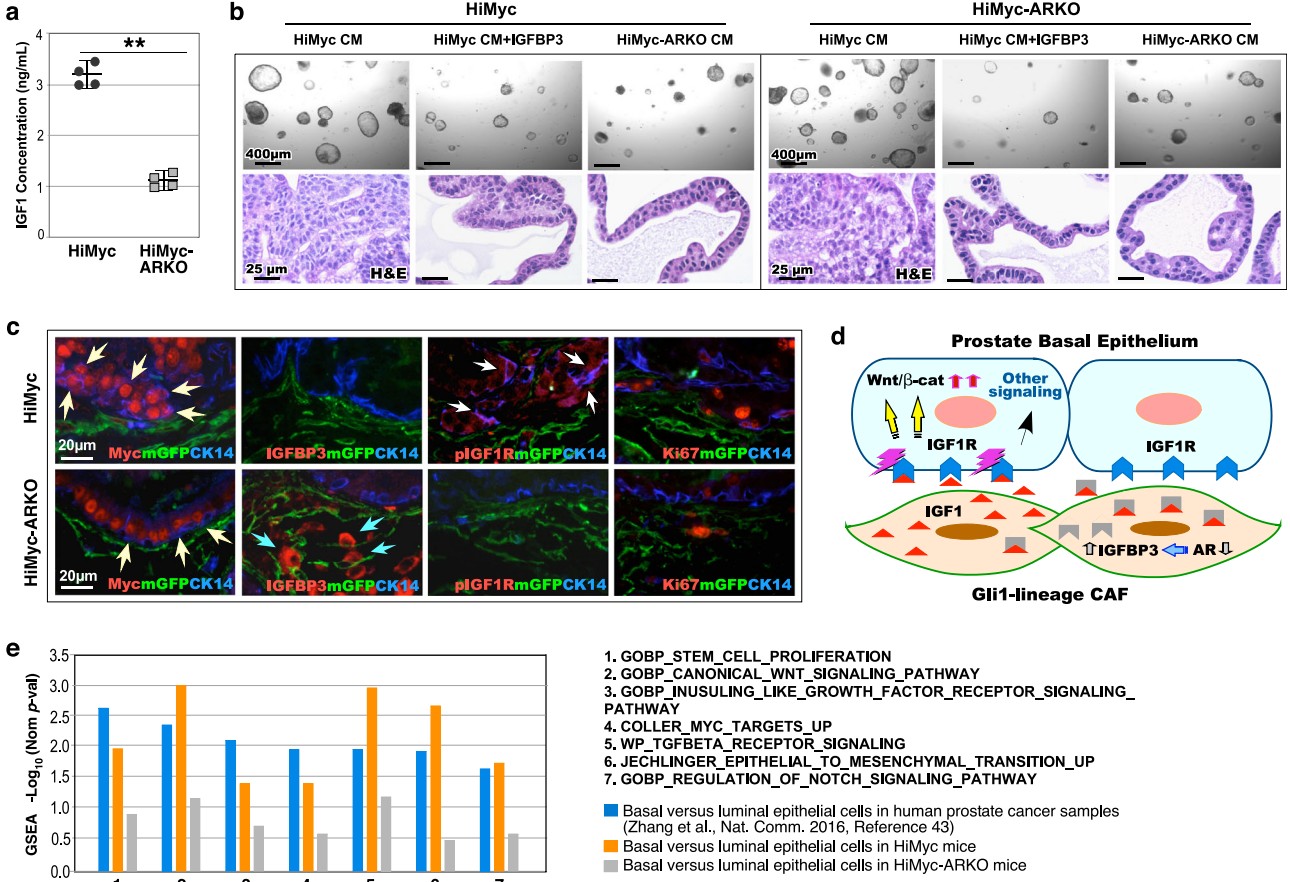

**Fig. 7 | Stromal AR in Gli1-lineage cells induces basal epithelial IGF and Wnt pathways to promote oncogenic growth through paracrine action. a** IGF1 concentration in conditioned media (CM) from 12-week-old *Hi-Myc:R26^{mTmG/+}: Gli1^{CreER/+}* and *Hi-Myc:R26^{mTmG/+}:Ar^{L/Y}:Gli1^{CreER/+}* GFP+ cells. Data are represented as mean ± SD of four independent samples over two biological replicates. Two-sided *t* test, **\*\***\*p* < 0.01. Source data and the exact *p* values are provided as a Source Data file. **b** Organoid culture of prostatic epithelial cells from 12-week-old *Hi-Myc:R26^{mTmG/+}: Gli1^{CreER/+}* and *Hi-Myc:R26^{mTmG/+}:Ar^{L/Y}:Gli1^{CreER/+}* mice, following treatment with *Hi-Myc:R26^{mTmG/+}:Gli1^{CreER/+}* CM, *Hi-Myc:R26^{mTmG/+}:Gli1^{CreER/+}* CM + IGFBP3 (1 µg/ml), or *Hi-Myc:R26^{mTmG/+}:Ar^{L/Y}:Gli1^{CreER/+}* CM. Representative brightfield images and HE staining showing organoid morphology and structure observed in indicated groups. Scale

bars, 400 µm and 25 µm. **c** Representative images of triple-IF staining for indicated antibodies on prostate tissues from *Hi-Myc:R26^{mTmG/+}:Gli1^{CreER/+}* and *Hi-Myc:R26^{mTmG/+}: Ar^{L/Y}:Gli1^{CreER/+}* mice. Yellow or white arrows indicate CK14+Myc+ or CK14+pIGF1R+ basal epithelial cells respectively. Blue arrows indicate IGFBP3+ mGFP+ stromal cells. Scale bars, 20 µm. **d** Schematic of hypothetic models by which stromal AR in Gli1-lineage cancer-associated fibroblast (CAF) cells regulate prostate tumorigenesis via IGF, Wnt, and other signaling pathways in prostatic basal epithelial cells. **e** GSEA results from pre-ranked differentially expressed genes list comparing basal cells versus luminal cells in human prostate cancer samples, *Hi-Myc:R26^{mTmG/+}: Gli1^{CreER/+}* and *Hi-Myc:R26^{mTmG/+}:Ar^{L/Y}:Gli1^{CreER/+}* mice. See also "Methods" section.

especially in AR-deficient fibroblasts in HiMyc-ARKO samples. Analyses of adjacent basal epithelial cells further showed a specific impairment of IGF1R signaling pathways in atypical basal epithelial cells in HiMyc-ARKO samples in comparison to HiMyc counterparts. Earlier studies have shown that the transcription of *Igfbp3* is mediated mainly through Sp1 transcription factor[35] and the AR can repress Sp1-mediated transcription by interfering with its binding to the target promoters[36,37]. Using ChIP assays, we further demonstrated increased recruitment of Sp1 within the *Igfbp3* promoter regions in sorted AR-deficient mGFP+ cells from HiMyc-ARKO mice in comparison to those of HiMyc counterparts. Because IGFBP3 is an IGF1 binding protein and can block IGF1 access to its receptors and inhibit IGF1 signaling activation[34], our data provide distinct mechanistic insight for stromal AR to support prostate basal epithelial cell oncogenesis through the regulation IGF1/IGFBP3 axes by reciprocal stromal-epithelial interactions.

Prostatic basal epithelial cells are directly adjacent to stromal cells and possess progenitor properties[23,25]. Specifically, Myc+ basal epithelial cells have been shown to function as prostate tumor-initiating cells in Hi-Myc mice[26]. Significant reduction of Myc+ atypical and tumor cells with basal cell properties was identified in prostatic tissues of HiMyc-ARKO mice. Accordingly, scRNA-seq analyses further showed

reduced Myc+ basal atypical and tumor epithelial cells but not Myc+ luminal counterparts in HiMyc-ARKO mice. Given the proximity of prostatic Gli1-lineage FB and basal cells, these data demonstrate a specific role for stromal AR in Gli1-lineage cells in regulating oncogenesis and tumor development initiated by prostatic basal epithelial cells. Interestingly, the expression of *Igf1* transgene in mouse prostatic basal cells initiates prostate oncogenesis and PIN development[46]. Activation of IGF1 axes further stabilizes cellular β-catenin and enhances its activity to directly promote prostate tumor development and growth[28,29,47]. In this study, we characterized the regulatory role of IGF1/IGF1R signaling in activating Wnt/β-catenin pathways in prostatic basal epithelial cells and promoting their oncogenic growth. Given that both IGF1 and Wnt/β-catenin signaling pathways have been shown to directly regulate androgen-induced prostate cancer development and progression[31], our current data implicate an underlying mechanism by which stromal AR regulates prostatic epithelial oncogenic transformation and tumor development.

In this study, we further examined the underlying mechanism for stromal AR in regulating prostatic basal epithelial oncogenesis using human clinical samples and data. To further address the clinical relevance of our analyses, we specifically selected human prostate cancer

specimens isolated either from primary prostate cancer patients who received no hormone treatment or from patients who only received Lupron treatment for 3 months in effort to reduce tumor volumes to achieve tumor-negative surgical margins. Positive staining for phosphorylated IGF1R only appeared in prostate tumor epithelial cells of untreated samples. In contrast, increased IGFBP3 staining and no phosphorylated IGF1R staining appeared in Lupron-treated samples (Supplementary Fig. 7a, b). These observations confirm the repressive role of androgen/AR signaling on IGFBP3 expression in human prostate stroma. To further explore the correlation of IGFBP3 status to IGF1 activation in human prostate cancer samples, we employed HiSeq RNA sequencing data from TCGA (The Cancer Genome Atlas) and GEO (GSE197780) datasets. IGFBP3 expression was significantly reduced in the naïve samples compared to the normal prostates ($p = 4.70 \times 10^{-6}$) and the ADT-treated samples ($p = 3.01 \times 10^{-7}$), suggesting a link between ADT and IGFBP3 expression. Moreover, GSEA using preranked gene lists of naïve versus ADT-treated samples showed significant enrichment in the IGF1 signaling pathway, indicating an inverse correlation between IGFBP3 expression and IGF1signaling activation (Supplementary Fig. 7c, d). We also observed similar results for increased IGFBP3 expression and reduced IGF1 activation using GSE197780 datasets, in which the patient samples were isolated after ADT in comparison to samples from the same patients before the treatment (Supplementary Fig. 7e, f). Whereas these data are supportive of our current results, we are also aware these samples contained both prostatic epithelial and stromal compartments. In order to fully understand the effect of ADT on stromal AR action in prostate cancer progression and CRPC development, more in-depth analyses should be carried out using clinically relevant clinical cohorts and datasets to assess cellular and molecular changes in both epithelial and stromal compartments.

Using organoid culture systems, we characterized the effect of stromal AR signaling in prostatic basal epithelial oncogenic growth. Interestingly, in earlier culture time points, we observed retarded growth in epithelial organoids derived from prostate basal cells of HiMyc-ARKO in comparison with those of HiMyc mice. However, the growth difference was gradually reduced at later time points. Additionally, IGF1 treatment of epithelial organoids derived from HiMyc and HiMyc-ARKO mice at late time points showed similar inducible growth. These data suggest that impaired epithelial tumor growth due to stromal AR loss in HiMyc-ARKO may be able to be reversed by microenvironment changes. Given that current ADT can systematically affect AR signaling in both AR-expressing tumor cells and surrounding stromal cells, our data implicate an underlying mechanism for IGF1 signaling forgoing the reliance on AR-mediated regulation in prostate tumor cells to induce hormone refractoriness and CRPC development. Therefore, further investigation of co-targeting reciprocal interactions of AR and IGF1 pathways between epithelial tumor cells and surrounding tumor niches may provide insight into potential therapeutic strategies for treating advanced prostate cancer.

In this study, we also tested the effect of androgen withdraw on both HiMyc and HiMyc-ARKO mice. We castrated 6-month-old HiMyc-ARKO and HiMyc mice that received TM at 2 weeks of age and analyzed them 4 weeks post castration (Supplementary Fig. 8). A regression of PIN and prostatic tumor growth was grossly and histologically apparent in both genotypes of mice. IHC analyses showed reduced Ki67 in PIN and prostatic tumor cells in castrated mice in comparison to intact counterparts of both genotypes of mice, indicating the promotional role of androgens in prostate epithelial growth. Interestingly, comparison of prostate weights before and after castration of both genotype mice showed greater reductions in prostate weight in HiMyc mice than HiMyc-ARKO mice, further suggesting the role of stromal AR in androgen-dependent prostate epithelial growth. The critical role of prostatic CAFs in tumor progression and CRPC development has been implicated recently[48]. Thus, further investigation of stromal AR action

in Gli1-lineage CAFs during ADT will help us better understand the molecular mechanisms underlying prostate cancer progression and hormone refractoriness, providing further insight into the development of potential and effective therapeutic strategies to co-target both tumor stromal and epithelial cells.

## Methods

### Ethics statement

All experimental procedures and care of animals in this study were carried out according to the Institutional Animal Care and Use Committee (IACUC) at Beckman Research Institute of City of Hope (California, US) and approved by the IACUC. Euthanasia was performed by $CO_2$ inhalation followed by cervical dislocation.

### Human subjects and clinical data

Specimens of primary prostate cancer used in this study were isolated from either patients without hormonal treatment or patients who received Lupron treatment for 3 months in reducing the tumor volumes to achieve tumor-negative surgical margins. The study was approved by the Institutional Review Board (IRB)-approved protocol (IRB # HS-16-00817) at the University of Southern California, and informed Consent was obtained from all participants in this study.

RNA-seq data for prostate adenocarcinoma patients were downloaded from TCGA database at UCSC Xena[49]. The gene expression was determined experimentally using the Illumina HiSeq 2000 RNA Sequencing platform by the University of North Carolina TCGA genome characterization center. Drug treatment status per sample was downloaded from the cBioportal website (http://www.cbioportal.org/). *IGFBP3* gene-levels were shown as in $\log_2(x+1)$ transformed RSEM normalized count. DEGs were considered using Wilcoxon Rank Sum test and a value of $|\log(\text{Fold Change})| > 0.1$ and adjusted $p$ values < 0.01. The GSE197780 dataset from the GEO database was used for validation.

### Mouse generation and experiments

All mice used in this study were from a C57BL/6 background, and housed in ventilated cage racks with free access to food and water under a 12 h light/dark cycle at 20–24 °C and 30–70% humidity at City of Hope Parvin Animal Facility. Mice containing the conditional *Ctnnb1* allele (*Ctnnb1*^L(ex3)^) were kindly gifted from Dr. Makoto M. Taketo[50]. *Pten*^L/L^ mice were kindly provided by Dr. Hong Wu[51], respectively. *PB-Cre4* (*PB*^Cre4^) mice were obtained from the NCI mouse repository (strain #: 01XF5). *Gli1*^CreER/+^ and *ROSA*^mTmG^ (*R26*^mTmG/+^) reporter mice were obtained from Jackson Laboratories (stocks 18867 and 7676). *Ar*^Lox/Y^ (*Ar*^L/Y^) and *Hi-Myc* transgenic (*Hi-Myc*) mice were kindly provided by Dr. Guido Verhoeven[52] and as reported earlier[18], respectively. *Ctnnb1*^L(ex3)/+^: *PB*^Cre4^ were generated by intercrossing *Ctnnb1*^L(ex3)/L(ex3)^ females with *PB*^Cre4^ male mice. *Pten*^L/+^: *PB*^Cre4^ mice were first generated by intercrossing *Pten*^L/L^ females with *PB*^Cre4^ males and then used to generate *Pten*^L/L^:*PB*^Cre4^. *Ar*^L/X^ female mice were mated with *Gli1*^CreER/+^ male mice to generate *Gli1*^CreER/+^ and *Ar*^L/Y^:*Gli1*^CreER/+^ littermates. Experimental mice were generated by intercrossing *Ar*^L/X^:*R26*^mTmG/+^:*Gli1*^CreER/+^ females with *Hi-Myc* males (Fig. 2a). Similar mating procedures were used to generate *Hi-Myc:R26*^mTmG/+^:*Gli1*^CreER/+^ and *Hi-Myc:R26*^mTmG/+^:*Ar*^L/Y^:*Gli1*^CreER/+^ mice (Supplementary Fig. 2a). Genomic DNA samples isolated from mouse tail tips or embryo yolk sacs were used for genotyping with appropriate primers (Ctnnb1 allele, forward: 5′-AACTGGCTTTTGGTGT CGGG-3′, reverse: 5′-TCGGTGGCTTGCTGATTATTTC-3′; Pten allele, forward: 5′-TCCCAGAGTTCATACCAGGA-3′, reverse: 5′-AATCTGTGCA TGAAGGGAAC-3′; PB^Cre4^ allele, forward: 5′-GATCCTGGCAATTTCGGC TAT-3′, reverse: 5′-GCAGGAAGCTACTCTGCACCTTG-3′; Gli1^CreER^ alleles, forward: 5′-GCAGATCTACATTCCTTTC-3′, reverse: 5′-AAGAGAGACA GCTGGAGCC-3′ and 5′-AATCGCGAACATCTTCAGGTT-3′; Ar alleles, forward: 5′-AGCCTGTATACTCAGTTGGGG-3′, reverse: 5′-AATGCATC ACATTAAGTTGATACC-3′; Hi-Myc alleles, forward: 5′-CAATGTCTGTG

TACAACTGCCAACTGGGATGC-3′, reverse: 5′-TTACGCACAAGAGTTCC GTAGCTGTTC-3′; *R26^mTmG/+* alleles, forward: 5′-TCAATGGGCGGGGG TCGTT-3′, reverse: 5′-CTCTGCTGCCTCCTGGCTTCT-3′ and 5′-CGAG GCGGATCACAAGCAATA-3′)[14,18,50,51]. For tissue recombination assays, timed pregnancy was done with *PB^Cre4* male mice mated with *Ctnnb1^L(ex3)/L(ex3)* female mice and *Pten^L/+*:*PB^Cre4* male mice mated with *Pten^L/L* female mice to generate urogenital sinus epithelium (UGE) tissues. *Gli1^CreER/+* male mice were mated with *Ar^L/X* female mice for timed pregnancy to generate UGM tissues. *Ar^L/X* Pregnant females were injected with TM (125 μg/g body weight, Sigma) suspended in corn oil (Sigma) at embryonic day 13.5 (E13.5) and euthanized on E16.5. The urogenital sinus (UGS) was separated into UGE and UGM by treatment with 1% trypsin (Gibco) at 4 °C for 90 min, followed by mechanical dissociation. Combinations of *Ctnnb1^L(ex3)/+*:*PB^Cre4* or *Pten^L/L*:*PB^Cre4* UGE and *Gli1^CreER/+* or *Ar^L/Y*:*Gli1^CreER/+* UGM were generated as indicated in Fig. 1a. The dissociated UGE and UGM were recombined on 0.4% agar plates containing Dulbecco's Modified Eagle's Medium (DMEM, Corning) with 10% fetal bovine serum (FBS, HyClone) supplementation, then incubated at 37 °C for overnight. The tissue recombinants were implanted under the renal capsule of 8-week-old male SCID mice with a supplement of testosterone pellet (12.5 mg, Innovative Research of America) placed in the back subcutaneously, and the grafts were analyzed 3 months later. All experimental mice received a single intraperitoneal injection of TM (1 mg) on postnatal day 14 (P14) or 125 μg/g body weight at 2 months of age for activation of *Gli1^CreER/+*. All mice-bearing tumors were closely monitored during the entire course of the study, and maximal tumor sizes (1.5 cm in diameter) were not exceeded based on the guidelines of IACUC in our institution.

## Pathological analyses and immunostaining

Mouse tissues were fixed in 10% neutral-buffered formalin (American Master Tech Scientific) and processed into paraffin. Following embedding in paraffin, tissue blocks were cut to 5 μm serial sections. For histological assessment, hematoxylin and eosin (H&E) staining[18,50,51] was performed and used in accordance with the guidelines recommended by The Mouse Models of Human Cancers Consortium Prostate Pathology Committee in 2013[16]. For immunohistochemistry (IHC), tissue sections were rehydrated through a decreasing ethanol gradient, treated by boiling in 0.01 M citrate buffer (pH 6.0) or Tris-EDTA (pH 9.0) for antigen retrieval, incubated in 0.3% $H_2O_2$ for 15 min, blocked by 5% normal goat/donkey serum (Gibco) in phosphate-buffered saline (PBS) for 1 h at room temperature, and incubated with indicated antibodies diluted in 1% normal goat/donkey serum at 4 °C overnight. Next, the slides were then incubated with biotinylated secondary antibodies for 1 h followed by horseradish peroxidase streptavidin (Vector Laboratories) for 30 min at room temperature and visualized using a DAB kit (Vector Laboratories). Counterstaining was performed with 5% (w/v) Harris Hematoxylin (Thermo Scientific) and dehydrated through an increasing ethanol gradient, and coverslips were mounted with Permount Medium (Fisher Scientific). For immunofluorescent (IF) staining, tissue sections were treated for antigen retrieval as described above, blocked in 5% normal goat/donkey serum for 1 h, and incubated with primary antibodies diluted in 1% normal goat/donkey serum at 4 °C overnight. Slides were washed in PBS then incubated with fluorescent-conjugated secondary antibodies for 1 h, and then mounted using VECTASHIELD Mounting Medium with DAPI (Vector Laboratories). For IF staining of sorted single cells, the same procedures as described above were used but replaced antigen retrieval step with a fixation step in 4% paraformaldehyde for 30 min and with blocking in 0.04% Triton X-100- 5% donkey serum in PBS. Both primary and secondary antibodies used in this study were listed in Supplementary Table 1.

## Preparation of dissociated prostate cells

Mouse prostate tissues were collected, minced into small pieces, and digested with 10 mg/mL collagenase (StemCell Technologies) in DMEM/Ham's F-12 50/50 Mix (DMEM/F12, Corning) supplemented with 10% FBS, 10 nM dihydrotestosterone (DHT, Sigma) and 10 μM Y-27632 (StemCell Technologies) at 37 °C for 90 min. Tissues were then digested with TrypLE (Gibco) supplemented with 1 nM DHT, 10 μM Y-27632 dihydrochloride at 37 °C for 15 min. Cell suspensions were passed through a 37-μm cell strainer (StemCell Technologies). Cell viability and concentration were detected using a TC Automated Cell Counter (Bio-Rad Laboratories) after Trypan blue (Gibco) staining, and cells at least 80% viability were processed. Similar experimental procedures were also described in the previous studies[21–24].

## Library preparation and single-cell RNA sequencing

Approximately 13,000 and 7,100 cells from 3-month-old *Hi-Myc:R26^mTmG/+*:*Gli1^CreER/+* and *Hi-Myc:R26^mTmG/+*:*Ar^L/Y*:*Gli1^CreER/+* littermates, respectively, were used for sequencing (Supplementary Fig. 2b). Library preparation was performed using 10× Genomics Chromium Single Cell 3′ Solution with v3 chemistry following the manufacturer's protocol (10× Genomics). Capillary electrophoresis using 2100 Bioanalyzer (Agilent Technologies) was used for validation of the library purity and size. The library quantity was measured using Qubit dsDNA HS Assay Kit (Invitrogen). cDNA libraries were sequenced on Illumina Novaseq 6000 S4 flow cell (Illumina) to a depth of 60–70 K reads per cell. Processing of raw sequencing data, including FASTQ file generation and Unique Molecular Identifiers (UMI) counting, was conducted using the 10× Genomics Cell Ranger pipeline (3.1.0). Reads were then aligned to the mm10 reference genome with added *mGFP* and *human c-Myc* transgene (*hMycTg*) sequences[18,53] for gene expression count. A total of 12,903 and 7,144 cells from *Hi-Myc:R26^mTmG/+*:*Gli1^CreER/+* and *Hi-Myc:R26^mTmG/+*:*Ar^L/Y*:*Gli1^CreER/+* prostate tissues, respectively, were uploaded, as filtered feature bar coded metrics, to R (4.2.1) using the Seurat package (3.2.1)[20] for the subsequent data analysis. For a quality-control step, potential empty droplets and multiplets were filtered out and dead cells or low-quality cells with a fraction of mitochondrial RNA higher than 15% were eliminated (Supplementary Fig. 2c, e, g, i). After this final filtering step, 10,810 *Hi-Myc:R26^mTmG/+*:*Gli1^CreER/+* cells with an average of 2,950 genes per cell and 15,736 UMI counts per cell, and 6,848 *Hi-Myc:R26^mTmG/+*:*Ar^L/Y*:*Gli1^CreER/+* cells with an average of 1,342 genes per cell and 5,041 UMI counts per cell were conserved for future analyses. For the UMAP visualization and analysis, normalized and scaled data were clustered using the top significant principal components of 5,000 highly variable genes with 0.5 resolution and 20 dimensions in Seurat (Supplementary Fig. 2d, f, h, j). Pathway analysis was performed using Gene Set Enrichment Analysis (GSEA 4.1.0). Spearman pairwise correlation matrices as the measure of association between IGF and Wnt axes genes were analyzed and plotted using the ggcorr function from the GGally package R (https://www.rdocumentation.org/packages/GGally/versions/1.5.0/topics/ggcorr) in R. Two individual sets of single-cell RNA sequencing experiments were performed using different littermates in this study.

## Cell sorting

Upon dissociation, cells were resuspended with 0.5% (w/v) Bovine serum albumin in PBS and sorted for mGFP+tdTomato- cells, or basal epithelial (CD24^low CD49f^high)[39,40]. All cell sorting experiments were carried out using an Aria Cell Sorter (BD Biosciences). After sorting, cells were dissolved in DMEM/F12 with 10% FBS and counted using Trypan blue. Purity of mGFP+ cells was confirmed by counting the number of mGFP+ cells compared to total number of cells stained negative for Trypan blue. Sorted prostatic basal cells were also validated by co-IF staining using antibodies against CK14, p63, and CK8 (Supplementary Fig. 5b, c). All of the samples used in the study possessed > 95% purity.

## RNA extraction and qRT-PCR

mRNA was extracted and purified from sorted cells using TRIZOL (Zymo Research) and reverse-transcribed using SuperScript IV First-

Strand Synthesis System (Fisher Scientific) according to the manufacturer's protocol. qRT-PCR reactions were performed in triplicate using SYBR Green PCR master mix (Applied Biosystems) with specific primers (Supplementary Table 2) on the 7500 Real-time PCR system (Thermo Fisher Scientific).

### Chromatin immunoprecipitation (ChIP)-qPCR

ChIP-DNA from sorted mGFP+ cells was obtained by ChIP assay. Briefly, cells were fixed with formaldehyde, cross-linked with glycine, and lysed in cold lysis buffer (0.2% SDS, 10 mM Tris-HCl [pH 8.0], 1 mM EDTA, and 0.5 mM EGTA). The chromatin was sheared in a range between 100–500 bp by sonication and diluted in ChIP dilution buffer (0.01% SDS, 167 mM NaCl, 16.7 mM Tris-HCl [pH 8.1], 1.1% Triton X-100, and 1.2 mM EDTA), and then subjected to immunoprecipitation by magnetic protein G beads (Invitrogen) conjugated with an AR antibody (Abcam, ab74272), SP1 antibody (Novus, NB600-233), or normal IgG (Cell Signaling) at 4 °C for 4 h. Crosslinks were reversed, and then chromatin DNA fragments were analyzed by real-time qPCR with specific primers (Supplementary Table 2).

### Conditional medium (CM) preparation

Following cell dissociation, mGFP+ cells were sorted and seeded in DMEM/Ham's F-12 50/50 Mix (DMEM/F12, Corning) supplemented with 10% FBS, 10 nM dihydrotestosterone (DHT, Sigma) and 10 μM Y-27632 (StemCell Technologies). After 12 h, culture media was then replaced with serum-free media after washing with PBS. The media was collected after 24 h for IGF1 concentration measurement and for treatment to organoids. IGF1 levels were measured in the above serum-free conditioned media using a Mouse/Rat IGF-1/IGF-1 Quantikine ELISA kit (R&D Systems). The samples were collected from two biologically different mice and measured in duplicate and tested and calculated following the manufacturer's protocol.

### Organoid culture

Following cell dissociation, cells were incubated in DMEM/F12 supplemented with 10% FBS and 10 nM DHT at 37 °C for 3 h. Basal epithelial cells (CD49f^high) were collected from suspended media, and approximately 2,000 cells per well were seeded in 20 μL of Matrigel (BD Biosciences) onto 24-well plates. Cells were cultured in serum-free DMEM/F12 containing 1× B27 (Life Technologies), 1.25 mM N-acetylcysteine (Sigma), 10 ng/mL EGF (PeproTech), 0.1 μg/mL Noggin (PeproTech), 0.1 μg/mL R-spondin1 (R&D Systems), 0.25 μM A83-01 (R&D Systems), 100 μM Y-27632, and 1 nM DHT[54]. After incubation for six days, organoids were treated with vehicle, 0.5% dimethyl sulfoxide (Sigma), 100 ng/mL insulin-like growth factor 1 (IGF1, R&D Systems), 100 ng/mL IGF1 + 1 μg/mL insulin-like growth factor binding protein 3 (IGFBP3, R&D Systems), IGF1 plus Wnt inhibitor ICG-001 5 μM (HY-14428, MedChem Express), IGF1 plus Wnt inhibitor iCRT3 5 μM (HY-103705, MedChem Express), *Hi-Myc:R26^{mTmG/+}:Gli1^{CreER/+}* CM only, *Hi-Myc:R26^{mTmG/+}:Gli1^{CreER/+}* CM + IGFBP3 1 μg/mL, or *Hi-Myc:R26^{mTmG/+}: Ar^{L/Y}:Gli1^{CreER/+}* CM only twice for four days and were fixed in 10% neutral-buffered formalin. Fixed cells were subjected to embedding into Histogel (Fisher Scientific) followed by paraffin embedding for histological analysis. All experiments were repeated three times in triplicates using three different mice. Organoid forming efficiency was determined by quantification of the percentage of organoid structure above 50 μm diameter per total cells seeded at day 0 in a well. The individual organoid size was quantified with the Image J (NIH) using at least 90 organoids per group. All experiments were replicated with two different mice in triplicate.

### Microscope image acquisition

H&E and IHC images were acquired on an Axio Lab A1 microscope using 10×, 20× and 40× Zeiss A-Plan objectives and captured using a Canon EOS 1000D camera and AxioVision software (Carl Zeiss, AxioVision Rel. 4.8 Ink.). Images of IF staining and organoids were taken using a Nikon ECLIPSE E800 epifluorescence microscope using ×5, ×20, and ×40 Nikon Plan Fluor objectives using a QImaging RETGA EXi camera and QCapture software (QImaging).

### Statistics and reproducibility

Data are presented as the mean values ± SD for the indicated number of independently performed experiments. The significance of the differences between data (*$p < 0.05$, **$p < 0.01$) was measured using two-sided $t$ test. Adjusted $p$ values were corrected for multiple testing using Benjamini–Hochberg's procedure. Differentially expressed gene (DEG) lists were determined using a Wilcoxon Rank Sum test, with genes showing adjusted $p$ value < 0.05 defined to be significant. Spearman's correlation coefficient > 0.3 and $p$ value < 0.05 were considered to indicate statistical significance. Enrichment scores (ES) for each gene set in the ranked list of genes were calculated by a running-sum statistic using GSEA. Nominal $p$ values of ES were estimated using an empirical phenotype-based permutation test and corrected for multiple hypothesis testing using FDR. As recommended by the GSEA User Guide, pathways with FDR < 0.25 were considered significant in exploratory GSEA pathway analysis. All representative images with consistent results from at least three biological replicates are shown.

### Reporting summary

Further information on research design is available in the Nature Research Reporting Summary linked to this article.

## Data availability

Raw data of single-cell RNA sequencing have been deposited in Gene Expression Omnibus (GEO) under accession number GSE174471. Human data and datasets were obtained from TCGA database, cBioportal, and from GSE 197780 dataset from GEO as detailed in the "Methods" section. Source data are provided with this paper.

## Code availability

The bioinformatics analyses were conducted using open-source software, including Cell Ranger version 3.1.0, Seurat version 3.2.1[20], R version 4.2.1, GSEA version 4.1.0. R scripts used to process sequencing data are available in "GitHub repository [https://github.com/wk-kim/HiMYC-ARKO-Gli1_Stromal_AR_Prostate_Tumorigenesis]"[55].

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

## Acknowledgements

This work was supported by NIH grants R01CA070297, R01CA166894, R01DK104941, and R01CA233664.

## Author contributions

A.H., W.K.K., and Z.S. conceived the project and designed the experiments. Y.H., A.H., A.P., W.K.K., D.H.L., C.H.N., A.J.B., and A.W.O. generated mouse colonies and performed genotyping and tissue collection experiments. A.H., W.K.K., Y.H., A.P., D.H.L., V.L., J.A., and Z.S. performed related mouse works and collected RNA, DNA, and single-cell samples for the analyses. A.H., W.K.K., A.P., and A.W.O. performed sequence experiments and analyzed sequencing data. A.H., W.K.K., D.H.L., G-Q. X., J.G., and Z.S. conducted ex vivo and in vivo experiments and performed staining and analyses. All authors were involved in analyses and confirmed data. A.H., W.K.K, A.W.O., and Z.S. wrote the manuscript.

## Competing interests

The authors declare no competing interests.
