## [Peer Review File · Nature Communications]

Stromal androgen signaling acts as tumor niches to drive prostatic basal epithelial progenitor-initiated oncogenesisREVIEWER COMMENTS

Reviewer #1 (Remarks to the Author):

Despite the tremendous clinical success of AR targeted therapies, majority of prostate cancer patients receiving those therapies will eventually develop resistance, which is one of the biggest challenges in managing this disease. Emerging evidence suggest the important roles of tumor microenvironment and stroma cells in mediating prostate cancer tumorigenesis and therapy resistance, while the detailed mechanism remain largely elusive. In this study, the authors take advantage of a series of novel GEMM models, especially the HiMyc-ARKO mice, which have AR-depletion specifically in the Gli1-lineage stroma cells, to examine the impact of AR depletion in stroma cells. Combing single cell sequencing and IHC/IF approaches, the authors showed that AR depletion led to ectopic expression of IGFBP3 in the FB cells, which bind and block IGF1/IGF1R signaling in adjacent basal prostate cancer cells and impair tumorigenesis. These results not only revealed a novel mechanism of how prostate stroma cells support the tumorigenesis and therapy resistance of PCa formation, but also suggest that co-targeting stromal and epithelial AR signaling as promising clinical revenue to combat AR therapy resistance.

Overall, I think this is a very promising and interesting study from both biological and technical points of view. The discovery of the key role of IGFBP3/IGF1/IGFR is not only compelling but may have strong clinical implication. Most of the experiments are well designed and support the conclusion. In summary, I think this manuscript may be a significant interest to the viewership of Nature Communication. However, I do have some concerns and questions related to this manuscript and I believe it will benefit from addressing them:

Major concerns:

1. Although the authors revealed detailed mechanism of how IGFBP3 was upregulated in the AR-depleted stroma cells which block the IGF1/IGF1R signaling in adjacent BE PCa cells, they did not show the exact mechanism how IGF1/IGF1R signaling regulate the AR and Myc signaling in BE PCa cells. How does decreased IGF1/IGF1R signaling impact the BE cell survival?
This is a specifically complicate question in the Hi-Myc model, as the impaired tumorigenesis in this model could be due to two reasons: 1) reduced AR signaling itself in BE PCa cells, or 2) reduced Myc signaling, which is actual oncogenic signal in this model. Interestingly, in Fig4h2, h4, k2 and k4, Myc expression was completely depleted in IGF1+IGFBP3 treated group, while AR is largely intact. What is the explanation of this?
2. One of the key conclusion and novelty of this study is indicating that the expression of IGFBP3 in stromal cell suppress the adjacent BE PCa cells through inhibiting their GF1/IGF1R signaling. However, the authors did not actually show that AR-depleted stromal cells indeed secrete reduced amount of IGF1 due to increased IGFBP3, which consequently suppress the IGF1/IGF1R signaling in adjacent BE PCa cells. The only related experiment presented here is treating organoid with IGF1 and IGFBP3, which didn't really prove this key point.
Alternatively, the authors should culture the sorted stromal cells, collect condition medium, and use those condition medium to treat the organoids and show that the condition medium from AR-depleted FB cells suppress IGF1/IGF1R signaling and organoid growth. Moreover, they should detect the amount of secreted IGF1 in the condition medium and show the reduced level of it. Finally, the authors should treat the condition medium from wt FB cells with IGFBP3 (to block the IGF1 in there), and then show that the condition medium would no longer support organoid growth (become similar as the AR-depleted condition medium).
3. The authors only provide very little human prostate cancer data in the end, which is quite weak to support the crucial conclusions. Additional clinical data, such as FFPE IHC showing that ectopic IGFBP3 signaling really occurs in the stroma of PCa treated with AR targeting therapies is important and a major gap at present. Furthermore, clinical correlation between IGFBP3/IGF1/IGF1R signaling with PCa progression should be examined using established patients' cohort, such as the SU2C and TCGA cohorts. Similarly, the authors could utilize some of the recent published patients single cell studies to validate their conclusions (exp: PMID: 33328604, 35058087).

Minor issues:

1. quantification was missing for some of the figures and conclusion. for example, line 73-74 stating "few cells showed positive B-catenin staining" but no quantification and statistics is presented.
2. Fig1d6 shows positive AR staining in some of the stromal cells (right up corner) but the authors stated AR staining is only in epithelial cells (line 75)?
3. some key related studies were not properly cited (exp: PMID: 32679108).
4. line 266, it looks like Fig4o should be Fig4n?

Reviewer #2 (Remarks to the Author):

The manuscript "STROMAL TUMOR NICHE DRIVES PROSTATIC BASAL EPITHELIAL PROGENITOR-MEDIATED ONCOGENESIS" describes a novel role of AR-signalling in prostate stroma cells and its impact on the epithelial compartment during prostate tumour development using very elegant mouse models. In particular, the authors report increased levels of IGFBP3 upon depletion of AR in stromal cells and their effect on the proliferation of HiMyc tumour cells.

The experimental procedures are very well described, defining a novel tumour non-autonomous mechanism of prostate tumorigenesis which is a potential novel mechanism of action of androgen deprivation therapies. However, the described mechanism does not really support co-targeting strategies to prostate cancer patients, as systemic androgen deprivation would concurrently decrease the levels of AR in the stromal and epithelial compartment. This needs to be discussed and corrected in the text.

Major concerns:

1. The authors described an upregulation of IGFBP3 and downregulation of IGF1 in stroma cells after depletion of AR at different timepoints, causing subsequent effects on epithelial cells. To really disentangle effects on stroma and epithelial cells (and to sustain their claim that co-targeting of both compartments is meaningful), it would be important to show effects of androgen deprivation (chemically or surgically) in their model of AR-depleted prostate stroma and subsequently on tumour development. In other words, what is the direct contribution of androgen-deprivation in epithelial cells?
2. Upon AR-loss in stromal compartment, a decreased in total number of basal cells is shown. Overall, an atypical CK14+ basal cell is described; however those cells can also be either intermediate cells or luminal progenitors. Co-stainings for CK14/CK8/p63 are required in mouse samples and FACS-sorted cells prior to organoid cultures. This point is critical, as the authors claimed that AR-loss induces in part an epithelial-transdifferentiation due to the presence of AR and K8 in organoid cultures.
3. The absence of AR in the stromal compartment may make extra androgen available for epithelial cells, which may result in supra-physiological androgen levels in situ and therefore promote cell differentiation. How is the AR-signalling/ signal-strength in basal and luminal cells from the HiMyc-ARko mice compared to HIMYC-ARwt mice?
4. Throughout the manuscript there are several references to the difference in cell numbers between HiMyc-ARko and HIMYC-ARwt mice, with only few real quantifications (such as Ki67, Myc, AR, CK14+). Further quantifications and statistical analysis are required to support those conclusions.
5. The conclusions that specific markers are co-expressed upon organoid-cultures (Figure 4) are not sustained by the presented data as the authors only provide individual stainings. Again quantification is required to support the conclusions that "more Myc+ atypical cells in IGF1-treated organoids than in IGF1+IGFBP3-treated..." is not quantified and may reflect technical variations.

Minor points:

- * The sorting strategy is unclear, as the gate in Ext. Fig5a is selecting for CD49f+CD24+, which enrich for basal and luminal cells. Therefore, it is not possible to conclude that the organoids are originated from atypical K14+ basal cells based on the current sorting strategy. The presence of AR and K8 probably just reflects the mixed population of origin rather than transdifferentiation. Analysis of the K14 and K8 content on the sorted cells would be key to support these statements. These statements also need to be corrected in the text.
- * The scRNAseq analysis shows some GFP expression in the epithelial compartment. How is the expression of Gli1 in epithelial cells, and in particular in basal cell subsets? Additionally, the expression of AR is surprisingly low in epithelial cells (Fig. 2c). Can the authors please clarify why this is the case?
- * Overall, the staining for K8 is poor, and it is difficult to conclude if the K14+ cells are basal cells or luminal progenitors/intermediate cells.

Reviewer #3 (Remarks to the Author):

This is a review for the manuscript entitled "Stromal Tumor Niche Drives Prostatic Basal Epithelial Progenitor-Mediated Oncogenesis".

The manuscript details a series of experiments carried out to determine the role of Androgen receptor signaling in stromal cells during prostate tumorigenesis. To achieve this, the authors mainly analyzed data from a prostate cancer mouse model, called the HiMyc model, where Myc expression is driven in the prostate using a synthetic chimeric promoter. To study Ar function in the stroma, they knocked out the Ar gene in the stromal cells (during development) using the Gli1-CreER allele and tamoxifen injection, and then assayed tumor phenotypes.

In terms of techniques, the authors analyzed these mice mainly through three techniques:

- 1) Tumor single-cell sequencing (about 1 mouse for each genotype, for a total of about 15,000 cells)
- 2) Imaging of tumor samples (mainly validating observations from the single-cell sequencing experiments)
- 3) Organoids (to determine the intrinsic/extrinsic effects of identified candidate regulator Igfbp3 on Igf1 signaling)

Altogether, the data provide a comprehensive picture with impactful conclusions for the treatment of this disease. It emerges that ArKO stromal cells express high levels of Igfbp3, an inhibitor of Igf1 signaling. Correlating with this, stromal-ArKO tumors show reduced Myc expression in Krt14+ basal-like cells, reduced Wnt signaling and reduced Igf1 signaling (Igf1r phosphorylation, CyclinD1 and Tcf7l2 levels). The causative link is then established through organoid cultures, where researchers show that the addition of Igf1 induces transformation features, whereas Igfbp3 blocks them... Myc staining of organoids validates the effects of Igfbp3 in the reduction of Myc, Krt14, and IGF1 signaling.

I think this is overall a very solid and robust study with plenty of interesting observations, which are generally well supported by results. The manuscript is, in certain parts, a bit disorganized, and I still feel like I do not understand the role of Figure 1 in the whole context of the paper. I also disagree with some conclusions and I believe that an understanding of the causes and consequences for the reduced numbers of Krt14+ cells in these tumors should be addressed with some additional focus. The manuscript would also be improved by checking spellings, verifying labels across all figures, and rephrasing some convoluted sentences.

My major concerns are the following:

- Is the PB-Myc model a good prostate cancer model for these studies? Is it possible that many of the effects observed are just the consequence of extrinsic signals regulating some unique features of the transgene, which is driven by a chimeric promoter?

- ArKO is verified only by IHC and RNA expression analysis, and I remain unconvinced that enough cells are KO. Is it possible to validate this by Western blot from sorted GFP+ cells?

- Tumors are analyzed at 3 months by single-cell, instead of waiting for 6-months for more aggressive tumors to develop. Is there any rationale for why the switch in timing from Figure 1 to Figure 2 experimental design?

- Based on images and single-cell data, the basal to luminal epithelial cell ratio seems really different in stromal-ArKO tumors... For me it is definitely the strongest phenotype. It almost seems like there are no basal-like cells in these tumors... or very few. I did not think this message came across as strongly in the text. Is this due to lack of basal cell expansion or is it driven by plasticity? Do these tumors have fewer tumor-reinitiating cells (per total cell number) when assayed by transplant? Are they less therapy-resistant?

- Based on the single-cell data, Myc expression is claimed to be lower in ArKO mice. However, UMI counts per cell in the ArKO sample are way lower than in the wt counterpart. Thus, the authors should consider random UMI subsampling, pre-normalization, to control for this. Alternatively, they should report $UMI(myc)/cell/cluster$ divided by $UMI(gfp)/cell/cluster$ as a better measure. I do agree in general that Myc is less expressed, but perhaps not to the extent that the authors claim it from the sc data.

- Similarly, I agree that Ar expression in stromal cells seems lower, as concluded by the authors, but normalization with GFP counts would look much nicer. Such large differences in UMI counts per cell can be misleading.

- Lack of replicates in single-cell experiments may also lead to some spurious correlations. Throughout the manuscript, it is critical that cherry-picked features from the single-cell dataset are validated by qPCR and IHC/IF across multiple independent mice.

- After all the mechanistic studies, it is still unclear to me why Myc levels go down in Krt14+ cells? How does Igf1 inhibition (through stromal signaling) actually end up impacting Myc expression?

- What happens if Ar is KO'd later, much later after tumors develop (at month 3 or 4, for example)... Can this also result in Myc downregulation? Do tumors stop growing? Do they lose cancer-reinitiating cells?

- Can recombinant Igfbp3 be added to tumor-model mice to see if this is sufficient to drive loss of Myc expression in the tumor?

Minor comments:

- "While comparable..." sentence in line 121 - I cannot understand its meaning.

- Some conclusions/statements pre-post reclustering the Epithelial cells seem repeated twice in the text. This makes reading a bit confusing to follow.

- CK8 and CK14 acronyms not explained correctly.

- Why Fig 2e shows Krt19 instead of Krt8?

- "However, ..." sentence in line 133 is too convoluted... please rephrase.

- "Therefore, identifying ..." sentence in line 142 is also convoluted... please rephrase.

- "Using triple-IF analyses..." in line 217... how do these experiments "directly" assess regulation between IGFBP3 and IGF1R signaling?

- Ki67 staining performed (line 224)... but no conclusions were made there or after.

- In line 240, the present tense is used to communicate results, incoherently with the rest of the text. Please address tense coherence throughout the manuscript.

- In line 253 the term "transdifferentiate" is used... wouldn't "differentiate" suffice... aren't CK14+ cells progenitors within the prostate? Then why "trans"? Any evidence of state plasticity being regulated by ArKO?

We greatly appreciate the Reviewers' time and effort to review this manuscript during the current pandemic time. Specifically, we thank them for their positive comments regarding the scientific significance, novelty, and clinical relevance of this study. We have taken each of the Reviewer's comments to heart and carefully address them with new and relevant experimental results. We also revised the related text in this revision appropriately and provide a point-by-point response to each of the Reviewers' comments below.

REVIEWER COMMENTS

Reviewer #1 (Remarks to the Author):

Despite the tremendous clinical success of AR targeted therapies, majority of prostate cancer patients receiving those therapies will eventually develop resistance, which is one of the biggest challenges in managing this disease. Emerging evidence suggest the important roles of tumor microenvironment and stroma cells in mediating prostate cancer tumorigenesis and therapy resistance, while the detailed mechanism remain largely elusive. In this study, the authors take advantage of a series of novel GEMM models, especially the HiMyc-ARKO mice, which have AR-depletion specifically in the Gli1-lineage stroma cells, to examine the impact of AR depletion in stroma cells. Combing single cell sequencing and IHC/IF approaches, the authors showed that AR depletion led to ectopic expression of IGFBP3 in the FB cells, which bind and block IGF1/IGF1R signaling in adjacent basal prostate cancer cells and impair tumorigenesis. These results not only revealed a novel mechanism of how prostate stroma cells support the tumorigenesis and therapy resistance of PCa formation, but also suggest that co-targeting stromal and epithelial AR signaling as promising clinical revenue to combat AR therapy resistance.

Overall, I think this is a very promising and interesting study from both biological and technical points of view. The discovery of the key role of IGFBP3/IGF1/IGFR is not only compelling but may have strong clinical implication. Most of the experiments are well designed and support the conclusion. In summary, I think this manuscript may be a significant interest to the viewership of Nature Communication. However, I do have some concerns and questions related to this manuscript and I believe it will benefit from addressing them:

We greatly appreciate the Reviewer's insightful comments on this study, including "...this is a very promising and interesting study from both biological and technical points of view. The discovery of the key role of IGFBP3/IGF1/IGFR is not only compelling but may have strong clinical implication....". We have carefully addressed each of his/her comments below.

Major concerns:

1. Although the authors revealed detailed mechanism of how IGFBP3 was upregulated in the AR-depleted stroma cells which block the IGF1/IGF1R signaling in adjacent BE PCa cells, they did not show the exact mechanism how IGF1/IGF1R signaling regulate the AR and Myc signaling in BE PCa cells. How does decreased IGF1/IGF1R signaling impact the BE cell survival?

We apologize for the confusion. In this study, we provided multiple lines of evidence to demonstrate the regulatory mechanisms by which activated IGF1/IGF1R axes induce Wnt/ β -catenin activation that further promotes prostatic basal epithelial oncogenic growth and tumor development. Results from our scRNA-seq analyses, qRT-qPCR, IHC, IF, and other experimental approaches have shown activated Wnt/ β -catenin signaling pathways in response to elevated IGF1/IGF1R signaling in prostatic basal epithelial cells in HiMyc mice in comparison to those in HiMycARKO counterparts. In this revision, we provided additional lines of scientific evidence to further strengthen the above findings. First, we showed the significant correlation among the expression of IGF1R and its downstream genes, as well as β -catenin and β -catenin-regulated target genes in hMyc+ BE cells of

HiMyc samples but not of HiMyc-ARKO samples (the current Fig.4c). Second, using triple-IF approaches, we provided mechanistic insight into activated IGF1R to induce the phosphorylation of AKT and GSK3 β , leading to increased stabilized β -catenin and the activation of its downstream targets in prostate BE cells of HiMyc samples (see Fig.4e). The above results were further validated using co-IF analyses (Supplementary Fig. 3k). Last, we showed the inhibition of Wnt signaling directly diminished IGF1 induced prostatic cell growth in epithelial organoid cultures derived from sorted BE cells (see Fig. 6e-f, Supplementary Fig. 6a-c). Specifically, the critical role of Wnt/ β -catenin signaling pathways in regulating Myc and AR-promoted prostate tumorigenesis has been well documented in the literature (Murillo-Garzon and Kypka, 2017; Yu et al., 2011; Zhu et al., 2004). Specifically, IGF1 and Wnt activation directly contributes to androgen-mediated prostate cancer development and progression (Kim et al., 2022). Our data and these lines of evidence fully support the regulatory role of IGF1 and Wnt/ β -catenin in prostate cancer growth and progression, and implicate a novel mechanism underlying stromal AR in regulating IGF1 and Wnt signaling to support prostate epithelial oncogenesis and tumor development through stroma-epithelium paracrine interactions. In this revision, we have provided these new and relevant data and modified the related texts.

This is a specifically complicate question in the Hi-Myc model, as the impaired tumorigenesis in this model could be due to two reasons: 1) reduced AR signaling itself in BE PCa cells, or 2) reduced Myc signaling, which is actual oncogenic signal in this model. Interestingly, in Fig4h2, h4, k2 and k4, Myc expression was completely depleted in IGF1+IGFBP3 treated group, while AR is largely intact. What is the explanation of this? ?

In order to address the impacts of stromal AR action on prostate epithelial tumorigenesis, we developed the current HiMyc-ARKO model, in which the expression of the *hMyc* transgene is regulated by a modified probasin promoter (ARR2PB) in prostatic epithelial cells and the *floxed* AR allele is regulated by the Gli1 promoter driven CreER in prostatic stromal cells. This mouse model allows us to specifically evaluate stromal AR action on Myc-induced prostatic epithelial tumorigenesis. As reported in this manuscript, we identified that stromal AR directly regulates IGF1 signaling through IGFBP3 in Gli1-lineage cells, which further induces Wnt/ β -catenin activation to promote prostate epithelial oncogenesis. To mimic the regulation between IGFBP3 and IGF1 *in vivo*, we pre-incubated IGFBP3 with IGF1 and then added them to prostatic organoid cultures. As shown in (previous Fig. 4 current Fig. 6a-d), fewer Myc⁺ cells, as well as reduced stabilized β -catenin, revealed in IGF1+IGFBP3 treated samples in comparison with IGF1-only treated samples, demonstrating the role of IGFBP3 in neutralizing IGF1 activity to reduce Wnt/ β -catenin signaling and Myc-mediated tumorigenesis. In response to the Reviewer, we provided quantified data for the above IHC experiments (Supplementary Fig. 5i) and revised the related text in the “Results” to make the above observations much more clear to the Reviewer.

2. One of the key conclusion and novelty of this study is indicating that the expression of IGFBP3 in stromal cell suppress the adjacent BE PCa cells through inhibiting their IGF1/IGF1R signaling. However, the authors did not actually show that AR-depleted stromal cells indeed secrete reduced amount of IGF1 due to increased IGFBP3, which consequently suppress the IGF1/IGF1R signaling in adjacent BE PCa cells. The only related experiment presented here is treating organoid with IGF1 and IGFBP3, which didn't really prove this key point. Alternatively, the authors should culture the sorted stromal cells, collect condition medium, and use those condition medium to treat the organoids and show that the condition medium from AR-depleted FB cells suppress IGF1/IGF1R signaling and organoid growth. Moreover, they should detect the amount of secreted IGF1 in the condition medium and show the reduced level of it. Finally, the authors should treat the condition medium from wt FB cells with IGFBP3 (to block the IGF1 in there), and then show that the condition medium would no longer support organoid growth (become similar as the AR-depleted condition medium).

To directly address the Reviewer's comments, we recently performed three sets of experiments. We first measured the level of secreted IGF1 in mGFP+ stromal cell-conditioned medium from either HiMyc or HiMyc-ARKO mice, and observed increased IGF1 levels in HiMyc samples (see Fig. 7a). Then, we tested the effect of the above conditional media on prostate epithelial organoid growth. Accordingly, mGFP+ stromal cell-conditioned medium from HiMyc mice showed an inducible role in prostatic epithelial organoid growth (Fig. 7b, Supplementary Fig. 6d-f). Finally, we assessed pre-incubation of the above conditioned medium with IGFBP3 and observed the reduced activity on organoid growth (Fig. 7b, Supplementary Fig. 6d-f). Altogether, these additional data are consistent with our other results and directly support the critical role of stromal cell-derived IGFBP3 in regulating IGF1/IGFR1 mediated prostate basal cell growth.

3. The authors only provide very little human prostate cancer data in the end, which is quite weak to support the crucial conclusions. Additional clinical data, such as FFPE IHC showing that ectopic IGFBP3 signaling really occurs in the stroma of PCa treated with AR targeting therapies is important and a major gap at present. Furthermore, clinical correlation between IGFBP3/IGF1/IGF1R signaling with PCa progression should be examined using established patients' cohort, such as the SU2C and TCGA cohorts. Similarly, the authors could utilize some of the recent published patients single cell studies to validate their conclusions (exp: PMID: 33328604, 35058087).

We greatly appreciate the Reviewer's point. During the entire course of the study, we have been extremely careful to design and perform the significant, relevant, and appropriate experiments and analyses to validate our findings with current human datasets. Because human prostatic cancer cells with basal cell gene signatures have been shown to link to advanced, metastatic, and castration-resistance tumor phenotypes (Zhang et al., 2016), we compared GSEA based on DEGs of basal versus luminal cells in human prostate tumors with the DEGs of basal cells from our mouse models. We observed very similar enriched signaling pathways between human samples and HiMyc mice, demonstrating the clinical relevance of stromal AR action as a tumor niche in supporting human prostate tumorigenesis. To directly respond to the Reviewer's comment, we recently obtained prostate cancer specimens isolated from both patients without hormonal treatment and patients who received Lupron treatment for 3 months to reduce the tumor volumes and achieve tumor-negative surgical margins. IHC analyses showed increased IGFBP3 expression in stromal cells and reduced phosphorylated IGF1R staining in adjacent basal epithelial cells of Lupeon treated samples in comparison to untreated samples (Supplementary Fig. 7). Additionally, we also assessed IGFBP3 expression and IGF1 signaling using current TCGA and other available human prostate datasets. We observed increased IGFBP3 expression and reduced IGF1 signaling in ADT treated samples in comparison with normal and untreated samples. Given the above datasets were generated from patient tumor tissues containing both epithelial and stromal cells, it is necessary to develop new and more relevant and independent clinical cohorts and datasets for conducting more in-depth analyses to examine the effect of ADT on stromal AR action in prostate cancer progression and CRPC development. Thus, results from the current study provide new insight into our understanding and need to be addressed further.

Minor issues:

1. quantification was missing for some of the figures and conclusion. for example, line 73-74 stating "few cells showed positive B-catenin staining" but no quantification and statistics is presented.

In response to the comment, we provided quantified analyses for the above IHC experiments in the current revision (see Supplementary Fig. 1a)

2. Fig1d6 shows positive AR staining in some of the stromal cells (right up corner) but the authors stated AR staining is only in epithelial cells (line 75)?

We have carefully examined the figures, and revised the related text in the revision (see lines 72-73)

3. Some key related studies were not properly cited (exp: PMID: 32679108).

We apologize for the confusion and have carefully checked the related citation (see text line 450, Ref. 48)

4. line 266, it looks like Fig4o should be Fig4n?

We have made the suggested changes in this revision.

Reviewer #2 (Remarks to the Author):

The manuscript “STROMAL TUMOR NICHE DRIVES PROSTATIC BASAL EPITHELIAL PROGENITOR-MEDIATED ONCOGENESIS” describes a novel role of AR-signalling in prostate stroma cells and its impact on the epithelial compartment during prostate tumour development using very elegant mouse models. In particular, the authors report increased levels of IGFBP3 upon depletion of AR in stromal cells and their effect on the proliferation of HiMyc tumour cells.

The experimental procedures are very well described, defining a novel tumour non-autonomous mechanism of prostate tumorigenesis which is a potential novel mechanism of action of androgen deprivation therapies. However, the described mechanism does not really support co-targeting strategies to prostate cancer patients, as systemic androgen deprivation would concurrently decrease the levels of AR in the stromal and epithelial compartment. This needs to be discussed and corrected in the text.

We greatly appreciate the Reviewer's positive comments regarding the model systems, experimental approaches, and the novel mechanism for ADT in this study. In particular, we thank his/her insightful question/comment on the future co-targeting therapeutic strategies. Our findings of identifying the repressive role of androgen/AR signaling in IGFBP3 expression in Gli1-lineage stromal cells elucidate a novel mechanism by which current ADT represses prostate epithelial tumor growth. Thus, dysregulation of AR-mediated repression on IGF1 in prostatic tumor stroma may forgo IGF1/IGF1R signaling to contribute to hormone refractoriness and CRPC development. A prominent role of IGF1/IGF1R signaling in promoting androgen-independent tumor growth has been identified *in patients and relevant in vivo models* (Biernacka et al., 2012; Mehta et al., 2011). Specifically, the regulatory role of IGFBP3 on prostate tumor progression and metastasis has also been reported (Mehta et al., 2011). Therefore, our data from this current study suggest co-targeting reciprocal interactions of AR and IGF1 pathways between epithelial tumor cells and surrounding tumor niches may improve clinical outcomes. In this revision, we have revised the related sections to make the above point explicitly to the Reviewers and readers (P4, the end of the first paragraph).

Major concerns:

1. The authors described an upregulation of IGFBP3 and downregulation of IGF1 in stroma cells after depletion of AR at different timepoints, causing subsequent effects on epithelial cells. To really disentangle effects on stroma and epithelial cells (and to sustain their claim that co-targeting of both compartments is meaningful), it would be important to show effects of androgen deprivation (chemically or surgically) in their model of AR-depleted prostate stroma and subsequently on tumour development. In other words, what is the direct contribution of androgen-deprivation in epithelial cells?

Again, we truly appreciate the Reviewer's insightful point. Our data of identifying a non-autonomous mechanism for stromal AR in Gli1-lineage cells tumor niches to promote prostate

epithelial oncogenesis and tumor development suggest a potential role of stromal AR+Gli1+ cells acting as tumor niches in tumor progression and hormone refractoriness through the course of ADT. To directly address the Reviewer's comments, we castrated a group of 6-month-old HiMyc-ARKO and HiMyc mice that received TM at the age of 2 weeks and analyzed them 4 weeks post castration. The regression of PIN and prostatic tumor growth appeared grossly and histologically in both genotypes of mice. IHC analyses further showed reduced Ki67 on PIN and prostatic tumor cells in castrated mice in comparison to intact counterparts. These data demonstrate the inhibitory role of ADT on prostatic epithelial cells in both HiMyc and HiMyc-ARKO mice. We have provided these additional data in Supplementary Fig. 8). Additionally, as discussed in this current revision, we fully agree that this significant and clinically relevant question should be fully addressed in the near future.

2. Upon AR-loss in stromal compartment, a decreased in total number of basal cells is shown. Overall, an atypical CK14+ basal cell is described; however those cells can also be either intermediate cells or luminal progenitors. Co-stainings for CK14/CK8/p63 are required in mouse samples and FACS-sorted cells prior to organoid cultures. This point is critical, as the authors claimed that AR-loss induces in part an epithelial-transdifferentiation due to the presence of AR and K8 in organoid cultures.

We appreciate the Reviewer's point. In this revision, we showed the overlay between CK8 and CK14 in Myc+ cells in HiMyc samples, suggesting their intermediate cell properties (Supplemental Fig. 3j1-4 and 3k1-4). Additionally, we also co-stained CK14/CK8, CK5/CK8, and p63/CK8 in prostate tissues of both genotype mice as the Reviewer suggested, and observed the overlay of CK14, CK5, and p63 with CK8 in HiMyc samples (please see Reviewer only Fig. 1c). Additionally, co-IF analyses of FACS-sorted cells also revealed CK8/CK14+ cells in HiMyc samples (Supplementary Fig. 5b). These data further suggest the critical role of stromal AR in driving prostate basal cell differentiation and growth.

3. The absence of AR in the stromal compartment may make extra androgen available for epithelial cells, which may result in supra-physiological androgen levels in situ and therefore promote cell differentiation. How is the AR-signalling/signal-strength in basal and luminal cells from the HiMyc-ARko mice compared to HIMYC-ARwt mice?

We greatly appreciate the Reviewer's suggestion. While AR deletion in stromal Gli1-lineage cells showed a significant biological effect and phenotypes in prostate tissues, there was no significant change of serum androgen levels in both HiMyc and HiMyc-ARKO mice (see Reviewers only Fig. 1a). In our *in vivo* tissue recombination assays, we observed different growth phenotypes of grafts in combination with ARKO and control UGMs in the same hosts that bear the exact same androgen levels (see Fig.1). Additionally, data from our sc-RNAseq analyses also did not show the significant differences of androgen downstream strength in both basal and luminal epithelial cells between HiMyc and Himyc-ARKO mice. Earlier studies have shown that stromal AR deletion in prepubescent Gli1-lineage cells can impair prostate epithelial development (Le et al., 2020), suggesting the important role of stromal AR in prostatic epithelial differentiation. Finally, as we mentioned above, we observed a similar regression of prostate epithelial growth for both castrated HiMyc and HiMyc-ARKO mice. The above lines of evidence suggest local androgen levels may not play a critical role in the above mouse models. We incorporated the above points in the "Discussion" in the response to the Reviewer's comment (see page 21).

4. Throughout the manuscript there are several references to the difference in cell numbers between HiMyc-ARko and HIMYC-ARwt mice, with only few real quantifications (such as Ki67, Myc, AR, CK14+). Further quantifications and statistical analysis are required to support those conclusions.

We apologize for the confusion. In the current revision, we provided additional quantifications and statistical analyses to support our scientific conclusions (please see Supplementary Fig. 1a, 1d, 1e;

Supplementary Fig. 4e, 4m; Supplementary Fig. 5i; Supplementary Fig. 6a-b, 6d-e, 6g; Supplementary Fig. 8d-e, etc). Additionally, we also provided a detailed source file that contains the original data and analyses as well as related Supplementary tables.

5. The conclusions that specific markers are co-expressed upon organoid-cultures (Figure 4) are not sustained by the presented data as the authors only provide individual stainings. Again quantification is required to support the conclusions that “more Myc+ atypical cells in IGF1-treated organoids than in IGF1+IGFBP3-treated...” is not quantified and may reflect technical variations.

In response to the Reviewer’s comment, we have provided the qualification analysis to assess Myc+ cells in the above experiments (Supplementary Fig. 5i).

Minor points:

** The sorting strategy is unclear, as the gate in Ext. Fig5a is selecting for CD49f+CD24+, which enrich for basal and luminal cells. Therefore, it is not possible to conclude that the organoids are originated from atypical K14+ basal cells based on the current sorting strategy. The presence of AR and K8 probably just reflects the mixed population of origin rather than transdifferentiation. Analysis of the K14 and K8 content on the sorted cells would be key to support these statements. These statements also need to be corrected in the text.*

We apologize for the confusion. In this study, we have followed the previous reports exactly for cell sorting experiments. It has been shown murine prostate CD24- CD49f^{high} cells possess basal epithelial cell properties (Karthaus et al., 2014; Lawson et al., 2007). We also analyzed sorted cells using co-IF assays. We observed 95 % of sorted basal cells showing CK14 and p63 positive staining. In response to the reviewer, we provided the above data in the current revision (please see Supplementary Fig. 5b-c).

** The scRNAseq analysis shows some GFP expression in the epithelial compartment. How is the expression of Gli1 in epithelial cells, and in particular in basal cell subsets? Additionally, the expression of AR is surprisingly low in epithelial cells (Fig. 2c). Can the authors please clarify why this is the case?*

The Reviewer raised a very important question regarding scRNA-seq analyses. Actually, it has been reported that cross-contamination has been observed in droplet-based single-cell RNA sequencing technologies, such as Chromium X, BD Rhapsody, and inDrop. Droplet-based scRNA sequencing requires mRNA in each cell to undergo lysis allowing for uniquely barcoded mRNA (Gao et al., 2020; Nieuwenhuis et al., 2020; Zheng et al., 2017). However, in the above technologies, ubiquitous contamination of ambient RNA during cell suspension has been observed in droplets (Yang et al., 2020; Young and Behjati, 2020). It has been suggested that the occurrence of ambient RNA cross-contamination can hinder gene expression of cell-type specific genes such as GFP. Our scRNA-seq data revealed low levels of mGFP expression in epithelial cells, between 0-1.5 per epithelial cell, in comparison with the stromal cells possessing 2-4 (log normalized data). However, all other validation experiments have shown no expression of mGFP proteins in epithelial cells in this study, which is also supported by other previous studies on the stromal cell properties of Gli1-lineage cells ((Le et al., 2020; Peng and Joyner, 2015; Peng et al., 2013).

** Overall, the staining for K8 is poor, and it is difficult to conclude if the K14+ cells are basal cells or luminal progenitors/intermediate cells.*

In response to the Reviewer’s comment, we performed additional co-IF experiments to define the cellular properties of intermediate cells (see Reviewer only Fig. 1c). As described in the study, we observed intermediate cells in HiMyc samples (see Figure 3g), suggesting the role of IGF1 and Wnt

signaling in driving cell differentiation during prostate tumorigenesis.

Reviewer #3 (Remarks to the Author):

This is a review for the manuscript entitled “Stromal Tumor Niche Drives Prostatic Basal Epithelial Progenitor-Mediated Oncogenesis”. The manuscript details a series of experiments carried out to determine the role of Androgen receptor signaling in stromal cells during prostate tumorigenesis. To achieve this, the authors mainly analyzed data from a prostate cancer mouse model, called the HiMyc model, where Myc expression is driven in the prostate using a synthetic chimeric promoter. To study Ar function in the stroma, they knocked out the Ar gene in the stromal cells (during development) using the Gli1-CreER allele and tamoxifen injection, and then assayed tumor phenotypes.

In terms of techniques, the authors analyzed these mice mainly through three techniques: 1) Tumor single-cell sequencing (about 1 mouse for each genotype, for a total of about 15,000 cells), 2) Imaging of tumor samples (mainly validating observations from the single-cell sequencing experiments), and 3) Organoids (to determine the intrinsic/extrinsic effects of identified candidate regulator Igfbp3 on Igf1 signaling)

Altogether, the data provide a comprehensive picture with impactful conclusions for the treatment of this disease. It emerges that ArKO stromal cells express high levels of Igfbp3, an inhibitor of Igf1 signaling. Correlating with this, stromal-ArKO tumors show reduced Myc expression in Krt14+ basal-like cells, reduced Wnt signaling and reduced Igf1 signaling (Igf1r phosphorylation, CyclinD1 and Tcf7l2 levels). The causative link is then established through organoid cultures, where researchers show that the addition of Igf1 induces transformation features, whereas Igfbp3 blocks them ... Myc staining of organoids validates the effects of Igfbp3 in the reduction of Myc, Krt14, and IGF1 signaling.

I think this is overall a very solid and robust study with plenty of interesting observations, which are generally well supported by results. The manuscript is, in certain parts, a bit disorganized, and I still feel like I do not understand the role of Figure 1 in the whole context of the paper. I also disagree with some conclusions and I believe that an understanding of the causes and consequences for the reduced numbers of Krt14+ cells in these tumors should be addressed with some additional focus. The manuscript would also be improved by checking spellings, verifying labels across all figures, and rephrasing some convoluted sentences.

We greatly appreciate the Reviewer’s comments regarding the significance and relevance of this study, such as “*Altogether, the data provide a comprehensive picture with impactful conclusions for the treatment of this disease.*” As shown in the previous Figure 1, we used both *in vivo* tissue recombination assays and newly developed genetically engineered mouse models to identify the regulatory role of stromal AR in Gli1-lineage cells in supporting prostatic epithelial oncogenesis and tumor development. In response to the Reviewer’s comment, we revised the previous Figure 1 and separated the *in vivo tissue recombination* and *mouse model* work to make the data more explicit to the Reviewers and readers. Additionally, we also provided new and additional experimental results and revised the related text sections to directly address the Reviewer’s other points below.

My major concerns are the following:

- Is the PB-Myc model a good prostate cancer model for these studies? Is it possible that many of the effects observed are just the consequence of extrinsic signals regulating some unique features of the transgene, which is driven by a chimeric promoter?

The HiMyc model referred above is one of the most common and useful mouse models that has been frequently used in the field of prostate cancer research. In HiMyc mice, the expression of the human *Myc* transgene is regulated by the modified probasin ARR2PB promoter and activates in mouse

prostate epithelial cells. Development of the both PIN and prostate invasive tumors has been observed in HiMyc mice (Ellwood-Yen et al., 2003). In this study, in order for us to evaluate intrinsic stromal AR signaling in regulating prostatic epithelial oncogenesis, we developed Hi-Myc:Ar^{L/Y}:Gli1^{CreER/+} mice, in which hMycTg expression in prostate epithelium co-occurs with stromal Ar deletion regulated by Gli1-driven CreER. In response to the Reviewer's comment, we revised the related texts to make the above rationale and biological relevance more clear in the revision (the first paragraph, page 6).

- ArKO is verified only by IHC and RNA expression analysis, and I remain unconvinced that enough cells are KO. Is it possible to validate this by Western blot from sorted GFP+ cells?

Actually, both *AR* floxed and *Gli1-CreER* alleles using in this study have been developed by other investigators for many years and frequently used in numerous studies in the field (De Gendt et al., 2004; Kerkhofs et al., 2009; Peng and Joyner, 2015; Peng et al., 2013). These mouse genetic tools have been frequently used in the field and proven to be reliable systems in *in vivo* studies. In this study, we demonstrated the specific effects of TM induced *CreER* activity in inducing mGFP expression and AR deletion in Gli1-lineage cells using many different and advanced experimental approaches. Specifically, we have shown reduced stromal AR expression in HiMyc-ARKO mice (Fig. 2f-g, Fig. 5d, Fig. 5g, and Supplementary Fig. 4c-d, etc). In direct response to the Reviewer's comment, we performed Western blot analyses using TM induced mGFP+ cells isolated from prostate tissues of both HiMyc and HiMyc-ARKO mice. Significant reduced AR expression was observed in HiMyc-ARKO samples (See Reviewer only Figure 1b).

- Tumors are analyzed at 3 months by single-cell, instead of waiting for 6-months for more aggressive tumors to develop. Is there any rationale for why the switch in timing from Figure 1 to Figure 2 experimental design?

We appreciate the Reviewer's comment. In this study, we observed high grade mouse PIN (3-4) lesions and prostatic adenocarcinoma lesions developed in 2- and 6-month-old *Hi-Myc:Gli1^{CreER/+}* mice, respectively. Therefore, to assess the transcriptomic changes during the course of PIN progressing to tumors, we isolated prostate tissues from 3-month-old *Hi-Myc:Gli1^{CreER/+}* and *Hi-Myc:Ar^{L/Y}:Gli1^{CreER/+}* mice for scRNAseq analyses. As detailed in this manuscript, based on the above experimental designs, we identified a new and significant role of stromal AR in regulating prostatic epithelial tumorigenesis through IGF1 and Wnt signaling pathways. In response to the Reviewer's comment, we revised the related text to make the scientific rationale more clear in this revision (see Line 120).

- Based on images and single-cell data, the basal to luminal epithelial cell ratio seems really different in stromal-ArKO tumors... For me it is definitely the strongest phenotype. It almost seems like there are no basal-like cells in these tumors... or very few. I did not think this message came across as strongly in the text. Is this due to lack of basal cell expansion or is it driven by plasticity? Do these tumors have fewer tumor-reinitiating cells (per total cell number) when assayed by transplant? Are they less therapy-resistant?

We appreciate this valuable point from the Reviewer. As shown in this study, we observed reduced atypical Myc+ basal cells in PIN tissues, as well as minor PIN and tumor lesions in HiMyc-ARKO mice. Given that prostatic basal epithelial cells are directly adjacent to stromal cells and possess progenitor properties, our data implicate a regulatory role of stromal AR in Gli1-lineage cells in prostate oncogenesis through prostatic basal epithelial cells. In response to the Reviewer's comment, we have carefully modified the related sections to make the above data explicitly in this revision (see Current Fig. 6 and 7, and supplemental Fig. 5).

- Based on the single-cell data, Myc expression is claimed to be lower in ArKO mice. However, UMI counts per cell in the ArKO sample are way lower than in the wt counterpart. Thus, the authors should consider random UMI subsampling, pre-normalization, to control for this. Alternatively, they should report $UMI(myc)/cell/cluster$ divided by $UMI(gfp)/cell/cluster$ as a better measure. I do agree in general that Myc is less expressed, but perhaps not to the extent that the authors claim it from the sc data.

We appreciate the reviewer's expert comments. The sc-RNAseq datasets presented in the current study were normalized and integrated as recommended by the Seurat standard workflow (PMC6687398). Specifically, through the analyses, we acknowledged the difference in UMI counts per cell between the HiMyc-ARKO and HiMyc samples. We normalized Myc expression using the methods similar to those the Reviewer suggested: $UMI(myc)/cell/BE$ cluster divided by $UMI(krt14)/cell/BE$ cluster per sample. Because *hMycTg* expression occurs in prostatic epithelial cells, we normalized Myc expression with Krt14 rather than mGFP+, Gli1-CreER induced stromal cell marker in the above mouse models. The normalized data revealed more than 1.6 times the Myc expression in prostate basal epithelial cells of HiMyc compared to HiMyc-ARKO samples.

		MYC-transgene	Krt14
HiMYCBE	UMI count	1936	210851
	Relative MYC UMI		0.0092
ARKOBE	UMI count	41	7237
	Relative MYC UMI		0.0057

		Ar	EGFP
HiMYCStro	UMI count	2028	42773
	Relative AR UMI		0.0474
ARKOStro	UMI count	618	26878
	Relative AR UMI		0.0230

- Similarly, I agree that Ar expression in stromal cells seems lower, as concluded by the authors, but normalization with GFP counts would look much nicer. Such large differences in UMI counts per cell can be misleading.

Similarly, in this study, we also normalized Ar expression with mGFP+ in stromal clusters, and observed the higher value of AR expression in HiMyc than ARKO sample, reflecting the effect of Gli1-CreER mediated AR deletion in stromal cells. In the current revision, we modified the related text in the "Materials and Methods" section to emphasize the above points.

- Lack of replicates in single-cell experiments may also lead to some spurious correlations. Throughout the manuscript, it is critical that cherry-picked features from the single-cell dataset are validated by qPCR and IHC/IF across multiple independent mice.

We apologize for the confusion. Actually, as indicated in the "Materials and Methods" section, we performed two individual sets of sc-RNAseq experiments using different littermates of mice in this study. Analyses from these two sets of sc-RNAseq showed very similar results. Thus, we presented detailed data from one of the two sets of analyses in this manuscript. Additionally, through the entire study, we have rigorously validated our data from scRNAseq analyses using relevant and technically challenging experimental approaches from multiple independent mouse samples. In response to the Reviewer's comments, we modified the related texts to make the above point more clear to the Reviewer and readers (P6, the last paragraph).

- After all the mechanistic studies, it is still unclear to me why Myc levels go down in Krt14+ cells? How does Igf1 inhibition (through stromal signaling) actually end up impacting Myc expression?

In this study, we demonstrate the regulatory role of stromal AR in activating IGF1 signaling that further elevates Wnt/ β -catenin signaling pathways to promote prostate basal epithelial oncogenesis and tumor development. In the current revision, we provided additional lines of evidence to further address our above findings (see Fig. 4, and Supplementary Fig. 6). Our current results implicate an underlying mechanism by which stromal AR action supports prostatic epithelial oncogenesis through the regulatory loop of AR-IGFBP3-IGF1-IGF1R-Wnt/ β -catenin-Myc signaling pathways between stroma-epithelium paracrine interactions.

- *What happens if Ar is KO'd later, much later after tumors develop (at month 3 or 4, for example)... Can this also result in Myc downregulation? Do tumors stop growing? Do they lose cancer-reinitiating cells?*

We appreciate this insightful point of the Reviewer. Actually, we did examine the effect of stromal AR in Gli1-lineage cells at a later time point in this study. As shown in the current Supplementary Fig. 1f (previous Extended Data Fig. 1d), we injected TM at 2-month-old *Hi-Myc:Ar^{L/Y}:Gli1^{CreER/+}* and *Hi-Myc:Gli1^{CreER/+}* mice and analyzed them at the age of 6 months. We observed impaired PIN and prostate tumor development in HiMyc-ARKO mice in comparison to age-matched HiMyc counterparts (Supplementary Fig. 1g-j). In response to the Reviewer's comment, we modified the related figures and texts to make the above results more clear to the Reviewer and readers.

- *Can recombinant Igfbp3 be added to tumor-model mice to see if this is sufficient to drive loss of Myc expression in the tumor?*

In this study, we observed a robust increase of IGFBP3 specifically in AR-deleted Gli1-lineage cells. Using organoid cultures, we further demonstrate the regulatory role of recombinant IGFBP3 in neutralizing IGF1 to inhibit IGF1/IGF1R promoted tumor cell growth. Because there are many IGFBP proteins, IGF growth factors, receptors, and other regulators that play a variety of different roles *in vivo*, we have not tested the effect of adding Igfbp3 systemically in order to focus on the specific role of IGFBP3 in prostate tissues in this study.

Minor comments:

- *"While comparable..." sentence in line 121 - I cannot understand its meaning.*

We apologize for the confusion and have rephrased this sentence and other related sentences in the current revision (lines 114-122).

- *Some conclusions/statements pre-post reclustering the Epithelial cells seem repeated twice in the text. This makes reading a bit confusing to follow.*

Again, we are sorry for the confusion. We have carefully gone through the entire manuscript and made all necessary changes.

- *CK8 and CK14 acronyms not explained correctly.*

We made them more clear as the Reviewer suggested.

- *Why Fig 2e shows Krt19 instead of Krt8?*

As the Reviewer suggested, we replaced the above plot with Krt8 (see Fig. 3c).

- *"However, ..." sentence in line 133 is too convoluted... please rephrase.*

We rephrased the sentence to make it more explicit (see lines 144-146)

- *"Therefore, identifying ..." sentence in line 142 is also convoluted... please rephrase.*

We revised this sentence as well (see lines 155-157).

- *“Using triple-IF analyses...” in line 217... how do these experiments “directly” assess regulation between IGFBP3 and IGF1R signaling?*

We revised this sentence in this current version (see lines 337-339).

- *Ki67 staining performed (line 224)... but no conclusions were made there or after.*

We appreciate this point and revised this sentence (see Lines 343-346).

- *In line 240, the present tense is used to communicate results, incoherently with the rest of the text. Please address tense coherence throughout the manuscript.*

We revised the above sentence and also addressed “tense” coherence throughout the manuscript.

- *In line 253 the term “transdifferentiate” is used... wouldn’t “differentiate” suffice... aren’t CK14+ cells progenitors within the prostate? Then why “trans”? Any evidence of state plasticity being regulated by ArKO?*

We seriously considered the Reviewer’s comment, and replaced “differentiate” from “transdifferentiate” in this revision (see Line 290).

Cited references for the point-by-point response to the Reviewers:

- Biernacka, K.M., Perks, C.M., and Holly, J.M. (2012). Role of the IGF axis in prostate cancer. *Minerva endocrinologica* 37, 173-185.
- De Gendt, K., Swinnen, J.V., Saunders, P.T., Schoonjans, L., Dewerchin, M., Devos, A., Tan, K., Atanassova, N., Claessens, F., Lecureuil, C., *et al.* (2004). A Sertoli cell-selective knockout of the androgen receptor causes spermatogenic arrest in meiosis. *Proc Natl Acad Sci U S A* 101, 1327-1332.
- Ellwood-Yen, K., Graeber, T.G., Wongvipat, J., Iruela-Arispe, M.L., Zhang, J., Matusik, R., Thomas, G.V., and Sawyers, C.L. (2003). Myc-driven murine prostate cancer shares molecular features with human prostate tumors. *Cancer Cell* 4, 223-238.
- Gao, C., Zhang, M., and Chen, L. (2020). The Comparison of Two Single-cell Sequencing Platforms: BD Rhapsody and 10x Genomics Chromium. *Current genomics* 21, 602-609.
- Karthaus, W.R., Iaquinta, P.J., Drost, J., Gracanin, A., van Boxtel, R., Wongvipat, J., Dowling, C.M., Gao, D., Begthel, H., Sachs, N., *et al.* (2014). Identification of multipotent luminal progenitor cells in human prostate organoid cultures. *Cell* 159, 163-175.
- Kerkhofs, S., Denayer, S., Haelens, A., and Claessens, F. (2009). Androgen receptor knockout and knock-in mouse models. *J Mol Endocrinol* 42, 11-17.
- Kim, W.K., Olson, A.W., Mi, J., Wang, J., Lee, D.H., Le, V., Hiroto, A., Aldahl, J., Nenninger, C.H., Buckley, A.J., *et al.* (2022). Aberrant androgen action in prostatic progenitor cells induces oncogenesis and tumor development through IGF1 and Wnt axes. *Nat Commun* 13, 4364.
- Lawson, D.A., Xin, L., Lukacs, R.U., Cheng, D., and Witte, O.N. (2007). Isolation and functional characterization of murine prostate stem cells. *Proc Natl Acad Sci U S A* 104, 181-186.
- Le, V., He, Y., Aldahl, J., Hooker, E., Yu, E.J., Olson, A., Kim, W.K., Lee, D.H., Wong, M., Sheng, R., *et al.* (2020). Loss of androgen signaling in mesenchymal sonic hedgehog responsive cells diminishes prostate development, growth, and regeneration. *PLoS Genet* 16, e1008588.
- Mehta, H.H., Gao, Q., Galet, C., Paharkova, V., Wan, J., Said, J., Sohn, J.J., Lawson, G., Cohen, P., Cobb, L.J., *et al.* (2011). IGFBP-3 is a metastasis suppression gene in prostate cancer. *Cancer Res* 71, 5154-5163.
- Murillo-Garzon, V., and Kypta, R. (2017). WNT signalling in prostate cancer. *Nat Rev Urol* 14, 683-696.
- Nieuwenhuis, T.O., Yang, S.Y., Verma, R.X., Pillalamarri, V., Arking, D.E., Rosenberg, A.Z., McCall, M.N., and Halushka, M.K. (2020). Consistent RNA sequencing contamination in GTEx and other data sets. *Nature communications* 11, 1933.
- Peng, Y.C., and Joyner, A.L. (2015). Hedgehog signaling in prostate epithelial-mesenchymal growth regulation. *Dev Biol* 400, 94-104.
- Peng, Y.C., Levine, C.M., Zahid, S., Wilson, E.L., and Joyner, A.L. (2013). Sonic hedgehog signals to multiple prostate stromal stem cells that replenish distinct stromal subtypes during regeneration. *Proc Natl Acad Sci U S A* 110, 20611-20616.
- Yang, S., Corbett, S.E., Koga, Y., Wang, Z., Johnson, W.E., Yajima, M., and Campbell, J.D. (2020). Decontamination of ambient RNA in single-cell RNA-seq with DecontX. *Genome biology* 21, 57.
- Young, M.D., and Behjati, S. (2020). SoupX removes ambient RNA contamination from droplet-based single-cell RNA sequencing data. *GigaScience* 9.
- Yu, X., Wang, Y., DeGraff, D.J., Wills, M.L., and Matusik, R.J. (2011). Wnt/beta-catenin activation promotes prostate tumor progression in a mouse model. *Oncogene* 30, 1868-1879.
- Zhang, D., Park, D., Zhong, Y., Lu, Y., Rycaj, K., Gong, S., Chen, X., Liu, X., Chao, H.P., Whitney, P., *et al.* (2016). Stem cell and neurogenic gene-expression profiles link prostate basal cells to aggressive prostate cancer. *Nat Commun* 7, 10798.

Zheng, G.X., Terry, J.M., Belgrader, P., Ryvkin, P., Bent, Z.W., Wilson, R., Zivaldo, S.B., Wheeler, T.D., McDermott, G.P., Zhu, J., *et al.* (2017). Massively parallel digital transcriptional profiling of single cells. *Nature communications* 8, 14049.

Zhu, H., Mazar, M., Kawano, Y., Walker, M.M., Leung, H.Y., Armstrong, K., Waxman, J., and Kypta, R.M. (2004). Analysis of Wnt gene expression in prostate cancer: mutual inhibition by WNT11 and the androgen receptor. *Cancer Res* 64, 7918-7926.

REVIEWERS' COMMENTS

Reviewer #1 (Remarks to the Author):

The authors have done exceptional work to revised the manuscript and addressed all my concerns. The manuscript is significantly improved.

Reviewer #2 (Remarks to the Author):

The revised manuscript "Stromal tumor niche drives prostatic basal epithelial Progenitor-mediated oncogenesis" demonstrate the role of stromal-AR signaling in prostate tumor growth and propose a potential AR-IGF1 co-targeting to target CRPC.

The authors addressed very nicely most of my concerns and have provided and enhanced supporting evidences to the proposed mechanism. I would really suggest including data on Reviewer only Fig 1c into the final manuscript. Although it has not been possible to address the role in vivo of AR-IGF1 co-targeting for tumour growth, the new statistical analysis provided robust evidences supporting its potential as a tentative therapies for prostate cancer patients.

Reviewer #3 (Remarks to the Author):

This manuscript "Stromal Tumor Niche Drives Prostatic Basal Epithelial Progenitor-Mediated Oncogenesis" has been substantially revised by the authors. Notably, many sentences and conclusions that were hard to interpret are now much easier to digest. The authors have successfully responded to my comments and, where appropriate, have toned-down conclusions and assigned them more specific effect sizes and confidence estimates. The more quantitative focus put on the revision is much appreciated, as many prior claims are now substantiated with the availability of raw data and statistical details. I will just add a few comments that I feel would add to the transparency and reproducibility:

- Authors should indicate n of samples in each case where a p-value is reported, most usefully in the figure legends.
- In the correlation tests with Igf1r and Wnt pathway targets, p-values for the significance of the correlation should be given in addition to r values.
- Using FB to abbreviate "fibroblast" adds too much confusion and too many abbreviations. Instead of using numbers to refer to clusters, it is much better to use one of their enriched markers to define them.
- In lines 230-234, an experiment analyzes AR occupancy on the Igfbp3 promoter to test whether AR-KO reduces this occupancy. If AR is being KO, what is the value of this experiment? Isn't all Ar occupancy everywhere expected to decrease when knocking-out its gene? I think this is a superfluous experiment. The changes in Sp1 recruitment should suffice. If the authors want to add anything, then Ar overexpression (or exogenous activation) could be tested to see if it is sufficient to displace Sp1 from the promoter and suppress Igfbp3 levels.
- In line 260, authors should make a clearer conclusion on the data obtained. Does this support that, as the subtitle implies, stromal AR signaling drives a CAF-like phenotype on some fibroblasts?
- Paragraph 297-311, the new results look great and add substantial impact to the mechanism. However, its writing needs some revision for both grammar and form, to be consistent with the rest of the article. In some sentences, like line 307, it is unclear what is being compared with what.
- Regarding the model proposed, can the IGF1 from CAFs also be secreted and block IGF1 coming from serum? Or is the mechanism purely through intrinsic blockade of IGF1 secretion?

Oct. 15, 2022

Reviewer #1 (Remarks to the Author):

The authors have done exceptional work to revised the manuscript and addressed all my concerns. The manuscript is significantly improved.

We greatly appreciate the Reviewer's time and effort to review our manuscript.

Reviewer #2 (Remarks to the Author):

The revised manuscript "Stromal tumor niche drives prostatic basal epithelial Progenitor-mediated oncogenesis" demonstrate the role of stromal-AR signaling in prostate tumor growth and propose a potential AR-IGF1 co-targeting to target CRPC.

The authors addressed very nicely most of my concerns and have provided and enhanced supporting evidences to the proposed mechanism. I would really suggest including data on Reviewer only Fig 1c into the final manuscript. Although it has not been possible to address the role in vivo of AR-IGF1 co-targeting for tumour growth, the new statistical analysis provided robust evidences supporting its potential as a tentative therapies for prostate cancer patients.

We appreciate the Reviewer's comments. Actually, we have provided several sets of experimental results to demonstrate the intermediate cell properties of Myc+ cells in both PIN and tumor lesions of HiMyc mice in the revision (see Fig. 3g and 3h, and Supplementary Fig. 3d).

Reviewer #3 (Remarks to the Author):

This manuscript "Stromal Tumor Niche Drives Prostatic Basal Epithelial Progenitor-Mediated Oncogenesis" has been substantially revised by the authors. Notably, many sentences and conclusions that were hard to interpret are now much easier to digest. The authors have successfully responded to my comments and, where appropriate, have toned-down conclusions and assigned them more specific effect sizes and confidence estimates. The more quantitative focus put on the revision is much appreciated, as many prior claims are now substantiated with the availability of raw data and statistical details. I will just add a few comments that I feel would add to the transparency and reproducibility:

- Authors should indicate n of samples in each case where a p-value is reported, most usefully in the figure legends.

In this revision, all data reporting p-values were indicated with the exact n of samples in the corresponding figure legends.

- In the correlation tests with Igf1r and Wnt pathway targets, p-values for the significance of the correlation should be given in addition to r values.

In response to the Reviewer's comment, we provided p-values for the correlation analysis between Igf1r, its downstream targets, or Wnt downstream targets in the revision (see Line 187-188).

- Using FB to abbreviate fibroblast adds too much confusion and too many abbreviations. Instead of using numbers to refer to clusters, it is much better to use one of their enriched markers to define them.

In this revision, we have made appropriate changes in response to the Reviewer's comments and following the Nat Commons guidelines for using abbreviation. Additionally, we also followed the previous publications to define the fibroblast (FB) clusters (Zhao *et al*, Single-cell transcriptome atlas of the human corpus cavernosum. *Nat Commun* **13**, 4302 (2022)).

- In lines 230-234, an experiment analyzes AR occupancy on the *Igfbp3* promoter to test whether AR-KO reduces this occupancy. If AR is being KO, what is the value of this experiment? Isn't all Ar occupancy everywhere expected to decrease when knocking-out its gene? I think this is a superfluous experiment. The changes in Sp1 recruitment should suffice. If the authors want to add anything, then Ar overexpression (or exogenous activation) could be tested to see if it is sufficient to displace Sp1 from the promoter and suppress *Igfbp3* levels.

Actually, the above data were included in the original submission but there were no concerns/questions raised in the previous review either by this Reviewer or others. Additionally, we provided the data in response to the previous reviewer's comment regarding if AR binds to the *Igfbp3* promoter in prostatic stromal cells. We do believe that these data directly address the previous Reviewer's question and also support the role of AR in interfering Sp1 on the *Igfbp3* promoter. Whereas we recognize the different scientific opinions between different Reviewers at different times, we also appreciate the consistent and meaningful review for our study.

- In line 260, authors should make a clearer conclusion on the data obtained. Does this support that, as the subtitle implies, stromal AR signaling drives a CAF-like phenotype on some fibroblasts?

Again, we appreciate the Reviewer's rigorous comment, and modified the related section in the current revision (see Line 273-274).

- Paragraph 297-311, the new results look great and add substantial impact to the mechanism. However, its writing needs some revision for both grammar and form, to be consistent with the rest of the article. In some sentences, like line 307, it is unclear what is being compared with what.

We appreciate the Reviewer's comment, and rephrased the related section and believe that the revised sentences are much clearer to the Reviewer and readers (see Line 315-323).

- Regarding the model proposed, can the IGFBP3 from CAFs also be secreted and block IGF1 coming from serum? Or is the mechanism purely through intrinsic blockade of IGF1 secretion?

This is very insightful point. Previous studies have shown IGFBP3 is an IGF1 binding protein in serum and thus can regulate IGF1 activity. As indicated in this revision, the important finding from our study is to identify the specific role of stromal AR in Gli1-lineage cells to support prostate basal epithelial cell oncogenesis through the regulation IGF1/IGFBP3 axes by reciprocal stromal-epithelial interactions.

Thank you again for handling this manuscript.

Sincerely yours

Zijie Sun, MD, PhD